# Variability of North Atlantic $CO_2$ fluxes for the 2000–2017 period estimated from atmospheric inverse analyses

Zhaohui Chen[1], Parvadha Suntharalingam[1], Andrew J. Watson[2], Ute Schuster[2], Jiang Zhu[3], and Ning Zeng[4]

[1]School of Environmental Sciences, University of East Anglia, Norwich, NR4 7TJ, UK.
[2]College of Life and Environmental Sciences, University of Exeter, Exeter, EX4 4RJ, UK.
[3]International Center for Climate and Environment Sciences, Institute of Atmospheric Physics, Chinese Academy of Science, Beijing,10029, China.
[4]Department of Atmospheric and Oceanic Science, and Earth System Science Interdisciplinary Center, University of Maryland, College Park, Maryland, 20742, USA

*Correspondence to*: Zhaohui Chen (Zhaohui.chen@uea.ac.uk)

**Abstract.** We present new estimates of the regional North Atlantic (15˚N–80˚N) $CO_2$ flux for the 2000–2017 period using atmospheric $CO_2$ measurements from the NOAA long term surface site network in combination with an atmospheric carbon cycle data assimilation system (GEOSChem–LETKF). We assess the sensitivity of flux estimates to alternative ocean $CO_2$ prior flux distributions and to the specification of uncertainties associated with ocean fluxes. We present a new scheme to characterize uncertainty in ocean prior fluxes, derived from a set of eight surface $pCO_2$–based ocean flux products, and which reflects uncertainties associated with measurement density and $pCO_2$–interpolation methods. This scheme provides improved model performance, in comparison to fixed prior uncertainty schemes, based on metrics of model−observation differences at the network of surface sites. Long term average posterior flux estimates for the 2000–2017 period from our GEOSChem–LETKF analyses are -0.255±0.037 PgC $y^{-1}$ for the subtropical basin (15˚N–50˚N), and -0.203±0.037 PgC $y^{-1}$ for the subpolar region (50˚N–80˚N, eastern boundary at 20˚E). Our basin–scale estimates of interannual variability (IAV) are 0.036±0.006 PgC $y^{-1}$ and 0.034±0.009 PgC $y^{-1}$ for subtropical and subpolar regions respectively. We find statistically significant trends in carbon uptake for the subtropical and subpolar North Atlantic of -0.064±0.007 and -0.063±0.008 PgC $y^{-1}$ decade$^{-1}$; these trends are of comparable magnitude to estimates from surface ocean $pCO_2$–based flux products, but larger, by a factor of 3-4, than trends estimated from global ocean biogeochemistry models.

## 1 Introduction

The ocean plays a key role in the global carbon budget, accounting for 2.5±0.6 PgC $y^{-1}$ of net $CO_2$ uptake from the atmosphere during the last decade (period 2009–2019), a level equivalent to ~26% of global fossil $CO_2$ emissions (Friedlingstein et al., 2020). The North Atlantic ocean has been identified a region of significant net oceanic $CO_2$ uptake in a range of recent analyses (Schuster et al., 2013, Landschützer et al., 2013, Lebehot et al., 2019), and also the location of the largest Northern Hemisphere uptake of anthropogenic $CO_2$ in recent decades (Gruber et al., 2019, Khatiwala et al., 2013, Sabine et al., 2004). Recent

estimates of net air–sea $CO_2$ fluxes derived from sea surface partial pressure $CO_2$ measurements ($pCO_2$) indicate net annual uptake for the North Atlantic over the past decade (2009–2018) with a range of 0.35–0.55 PgC $y^{-1}$ (Landschutzer et al., 2016; Rodenbeck et al., 2013; Zeng et al., 2015; Watson et al., 2020), and equivalent to about 14%–22% of the global net ocean carbon ocean sink reported for this period. Regionally aggregated air–sea $CO_2$ fluxes over the North Atlantic basin also display significant variability on interannual (Watson et al., 2009) and decadal timescales (Landschützer et al., 2016, 2019). Based on analyses of surface $pCO_2$ measurements, variations in regional $pCO_2$ trends were observed in the subtropical and subpolar regions, potentially associated with large–scale climate oscillations such as the North Atlantic Oscillation and the Atlantic Multi–decadal Variation (McKinley et al., 2011, Landschützer et al., 2019, Macovei et al., 2020). Devries et al. (2019) estimated a negative trend (i.e., a strengthening ocean sink) in North Atlantic $CO_2$ uptake for the 2000–2009 period based on analysis of $pCO_2$–based estimates and ocean models. Lebehot et al. (2019) find statistically significant trends in surface ocean $CO_2$ fugacity ($fCO_2$) for the 1992–2014 period from both observation–based surface mapping methods and from the CMIP5 Earth System models.

Recent analyses of North Atlantic air–sea $CO_2$ fluxes have primarily been based on 'bottom–up' methods of varying complexity which use interpolated surface ocean $pCO_2$ distributions (derived from in situ $pCO_2$ measurements) in combination with parameterizations of air–sea gas exchange (e.g., Landschützer et al., 2013; Rödenbeck et al., 2015; Takahashi et al., 2002, 2009). Estimates of air–sea $CO_2$ fluxes have also been derived by alternative methods such as global ocean biogeochemical models (e.g., Buitenhuis et al., 2013; Law et al., 2017), and 'top–down' methods which involve the application of inverse analyses or data assimilation methods to atmospheric and oceanic $CO_2$ measurements (e.g., Gruber et al., 2009, Mikaloff–Fletcher et al., 2006, Gurney et al., 2003, Peylin et al., 2013). 'Top-down' analyses estimate surface $CO_2$ fluxes by using information on observed gradients in atmospheric $CO_2$ together with atmospheric transport constraints (typically from 3 D atmospheric models) and prior information on the magnitude and associated uncertainties of surface $CO_2$ flux distributions (Rödenbeck et al., 2003; van der Laan–Luijkx et al., 2017; Peters et al., 2005; Peylin et al., 2013; Chevallier et al., 2014; Gaubert et al., 2019).

Previous studies also note that estimates of carbon fluxes from the atmospheric inverse method are sensitive to the specification of the prior flux distribution and its associated uncertainty distribution (Carouge et al., 2010; Chatterjee et al., 2013; Peylin et al., 2013). While there have been recent studies evaluating the sensitivity of land–based carbon flux estimates to specification of the prior flux and its uncertainty, there has been far less examination of ocean flux estimates from inverse methods. Several global inverse model assessments of the past decade have relied on the climatological ocean–atmosphere $CO_2$ flux database of Takahashi et al. (2009) to specify prior ocean fluxes. In view of the limited information available on the temporal and spatial variability of ocean carbon fluxes from this climatological ocean database, these inverse analyses have adopted different approaches to the specification of prior uncertainty for ocean fluxes, ranging from uncertainties derived from a separate ocean model inversion (in the case of Nassar et al., 2011), to a specified percentage of the prior flux magnitude (Feng et al., 2016, Liu et al., 2016).

In this study we present a new long term estimate of North Atlantic air–sea $CO_2$ fluxes for recent decades (period 2000–2017) using atmospheric inverse methods. We focus in particular on the specification of prior ocean fluxes (including sensitivity of flux estimates to alternative prior flux distributions) and on their associated flux uncertainties. To our knowledge these influences on inverse estimates of North Atlantic $CO_2$ flux have not been assessed previously. We use the carbon cycle data assimilation system GEOSChem–LETKF (GCL, described further in Sect. 2) which combines the global atmospheric $CO_2$

transport model GEOS–Chem (Nassar et al., 2010) with the Local Ensemble Transform Kalman Filter (LETKF) data assimilation system (Hunt et al., 2007; Miyoshi et al., 2007; Liu et al., 2019). In recent years several new global air–sea $CO_2$ flux products have been developed based on mappings of ocean surface $pCO_2$ measurements (e.g., Landschutzer et al., 2016, Rodenbeck et al., 2014, Watson et al., 2020, and products reported in the intercomparison of Roedenbeck et al., 2015). These ocean flux distributions are frequently derived from interpolations of surface ocean $pCO_2$ measurements from the SOCAT

database (Bakker et al., 2016) together with parameterizations of air–sea gas exchange. Following recent updates, the surface ocean $pCO_2$ database SOCATv2020 (https://www.socat.info/index.php/data-access/), now includes over 28 million surface ocean carbon measurements. The SOCAT database provides a valuable resource towards the development of bottom–up estimates of ocean–atmosphere $CO_2$ fluxes, and a compilation of these flux products is reported in the recent Global Carbon Budget (Friedlingstein et al., 2020). The increased range of global air–sea $CO_2$ flux products available (beyond the Takahashi

et al., 2009 climatology) provides a valuable opportunity to develop an improved representation of air–sea $CO_2$ flux variability and a more robust characterization of the uncertainties associated with ocean carbon fluxes. In this study we employ some of the recently developed ocean $CO_2$ flux products to provide a new method of characterizing the prior ocean flux uncertainty used for atmospheric inverse analyses. The methodology is based on the ensemble spread of the multiple ocean flux products, and reflects underlying uncertainties in these products, such as those associated with sampling density of the surface

measurements and interpolation method employed. It provides a spatially and temporally variable specification of prior flux uncertainty that will be of value to the inverse modeling community.

The remainder of the paper is organized as follows: Section 2 covers the methodology of the atmospheric inverse analysis, outlining the carbon cycle data assimilation system (GEOSChem–LETKF), the atmospheric $CO_2$ observations, and specifications of prior fluxes and uncertainties. Further details of the methodology are presented in the Appendix. In Sect. 3

we present GEOSChem–LETKF assessments of alternative specifications of ocean prior fluxes and flux uncertainties, and then use these results to derive long term estimates of North Atlantic $CO_2$ fluxes for the 2000–2017 period. We also summarize specific characteristics of North Atlantic $CO_2$ fluxes derived from these analyses, namely, long term means, trends, and interannual variability of fluxes, and compare our results with other recent relevant studies.

## 2 Materials and Methods

### 2.1 Overview

Our analysis employs the global GEOS–Chem atmospheric chemistry transport model together with the Local Ensemble Transform Kalman Filter (LETKF) (described in Sect. 2.2) and atmospheric $CO_2$ observations from the NOAA–ESRL network of surface sites (Sect. 2.3). Section 2.4 describes the compilation of the set of air–sea $CO_2$ flux products and the derivation of the prior flux uncertainty specification for the North Atlantic based on the ensemble spread of these products. Section 3 presents model results, including sensitivity analyses assessing different prior flux representations and flux uncertainty schemes (Sect. 3.1), and regional $CO_2$ flux estimates for the 2000–2017 period from the GEOSChem–LETKF system (Sect. 3.2). Further details on model analyses, observations and uncertainty calculations are presented in the sections below and in the Appendix.

### 2.2 The GEOSChem−Local Ensemble Transform Kalman Filter (GCL) system

The GEOS–Chem atmospheric chemistry transport model has been used in a range of previous investigations into atmospheric $CO_2$ and applied in conjunction with inverse analyses to estimate surface carbon fluxes (Nassar et al., 2010, 2011; Suntharalingam et al., 2005; Liu et al., 2016). In this analysis we employ GEOSChem v11–01 at a horizontal resolution of 2° latitude by 2.5° longitude, with 47 levels in the vertical. Model transport fields are provided by GEOS–5 assimilated meteorological data from the NASA Global Modeling and Assimilation Office (GMAO, Rienecker et al., 2008). The GEOSChem configuration employed here primarily follows that of Nassar et al. (2011), but with updated representation of prior fluxes; more detail on the prior $CO_2$ fluxes and uncertainties implemented in this study is given in Sect. 2.4.

The Local Ensemble Transform Kalman Filter (LETKF) is a data assimilation system which provides an estimate given a prior (or "background") estimate of the current state based on past and current data (in this case, the atmospheric $CO_2$ mole fraction observations). The general framework of the LETKF is described in Hunt et al. (2007); it has been adapted by Miyoshi et al. (2007) to provide gridscale localized analysis of flux estimates. The LETKF system has been used to estimate $CO_2$ fluxes in a range of previous studies (e.g, Kang et al., 2012; Liu et al., 2016, 2019). The LETKF provides iterative estimates of the time evolution of the system state, $x$, (here representing the grid–scale surface carbon fluxes). Each step involves a forecast stage (based on a physical model of the system evolution) and a state estimation stage (the 'analysis' step), which combines system observations, $y$, together with the background forecast, $x^b$, to derive the improved state estimate. The observation operator $H$ provides the mapping from the state space to the observation space; in this study $H$ is provided by the GEOS–Chem atmospheric model.

In this analysis we employ the complete GEOSChem–LETKF (GCL) data assimilation system to conduct sensitivity analyses on the ocean prior fluxes, and to provide a long term flux estimate of surface $CO_2$ fluxes for the North Atlantic for the period 2000–2017. We report a posteriori fluxes on monthly timescales for the 2000–2017 period; the optimized monthly fluxes are derived from four sequential weeks of assimilation cycles, as further described below. Our methods follow the implementation of the LETKF system by Liu et al. (2019), who have extended the previous carbon data assimilation system of Kang et al.

(2011, 2012). The study of Kang et al. (2011) assimilated meteorological data and atmospheric $CO_2$ concentrations to provide estimated atmospheric $CO_2$ concentrations as part of the state estimate. Kang et al. (2012) extended this method to also provide estimates of surface carbon fluxes. Both these LETKF studies assimilated meteorological data and atmospheric $CO_2$ concentrations and employed a short assimilation window of 6 hours in order to maintain linear behaviour of the ensemble perturbations (Kang et al., 2011, 2012). In addition, Kang et al. (2012) also tested longer assimilation windows (up to 3 weeks) for LETKF formulations that assimilated atmospheric $CO_2$ concentrations alone (eliminating the assimilation of the meteorological data). The LETKF system of Liu et al. (2019) extended the Kang et al. (2011, 2012) analyses by incorporating the GEOSChem atmospheric model as the forecast model, along with its representation of surface $CO_2$ fluxes which provide the prior flux specification for the forecast step. However, Liu et al. (2019) assimilate only atmospheric $CO_2$ measurements (i.e., no assimilation of meteorological measurements), and use an assimilation window of 7 days; the duration of the assimilation window was selected to maximize the correlation between observations and surface fluxes. The GEOSChem–LETKF system employed in our study follows the Liu et al. (2019) formulation; atmospheric $CO_2$ measurements are assimilated at 7 day timescales, with the LETKF analysis step providing updates of the surface fluxes and associated uncertainties required as initial conditions for the next weekly forecast step. We report monthly flux estimates following four assimilation cycles. Further details on the LETKF and the governing equations for flux estimation are provided in Appendix A.

## 2.3 Atmospheric CO₂ Observations

Atmospheric $CO_2$ observations used for this study are taken from the NOAA–ESRL GLOBALVIEWplus Observation Package v4.2 (Obspack, Cooperative Global Atmospheric Data Integration Project, 2018). $CO_2$ measurement records for the period 2000–2017 from 86 surface sites were used in this analysis. Further details on the measurement sites and the site–specific observation uncertainty characteristics are presented in Table A1 of the Appendix. The specification of observational uncertainty associated with incorporation of the atmospheric $CO_2$ measurements into the LETKF is derived using the methods of Chevallier et al. (2010); we use the standard deviation of measurement variability from detrended and deseasonalized $CO_2$ time series at each measurement site. The resulting specification of observational uncertainty varies between 0.16 ppm (for stations in and around the Southern Ocean) to over 5 ppm (for stations in continental interiors) (see Appendix Table A1 for more details).

## 2.4 Specification of Prior CO₂ Fluxes and Associated Flux Uncertainties

The GEOSChem model $CO_2$ simulation employed in this study includes representation of fossil fuel emissions, air–sea fluxes and exchange with the terrestrial biosphere. Details of the data sources used to specify the prior flux distributions are outlined here. Fossil fuel emissions are taken from Chevallier et al. (2019) (Global Atmospheric Research version 4.3.2, Crippa et al.,

2016, scaled globally and annually from Le Quéré et al., 2018), and land biosphere fluxes from the Joint UK Land Environment Simulator (JULES, Clark et al., 2011).

The focus of our study is on North Atlantic Ocean $CO_2$ fluxes, and we investigate the representation of ocean prior fluxes and prior flux uncertainty in more detail. Firstly, in Sect. 3.1, in a set of sensitivity analyses, we compare the implementation of three different representations of ocean $CO_2$ fluxes that have been used to specify prior fluxes in recent inverse analyses: (i) the widely used Takahashi et al. (2009) climatology (hereinafter Ta), (ii) the interannually varying flux product of Landschützer et al. (2016) derived from surface $pCO_2$ distributions (hereinafter La), and (iii) the interannual fluxes from the ocean mixed-layer scheme of Rödenbeck et al. (2014) (hereinafter Ro). We also evaluate in more detail, the impact of different specifications of prior flux uncertainty for ocean fluxes. Many previous atmospheric inverse estimates of air–sea carbon fluxes have employed relatively simple characterizations of the prior ocean flux uncertainty, for example, based on a fixed proportion of the grid–scale or regional prior flux (Nassar et al., 2011, Liu et al., 2016, Feng et al., 2016). In Sect. 3.1, we employ both fixed flux uncertainties, and also present an alternative scheme derived from the ensemble spread of ocean $CO_2$ flux products, as described below.

The prior ocean flux distributions employed in atmospheric inversions are frequently derived from interpolations of the surface ocean $pCO_2$ database (e.g., SOCAT, Bakker et al., 2016) in combination with ocean–atmosphere gas exchange parameterizations. Uncertainties in the derived products stem from uncertainties in the input data (e.g., density of measurements), interpolation methods, and gas–transfer parameterizations (Landschutzer et al., 2013). However, some ocean regions, the North Atlantic in particular, have a higher density of $pCO_2$ measurements and more consistent flux estimates from $pCO_2$–based products (Schuster et al., 2013, Landschutzer et al., 2013). Here we exploit the recent expansion of $pCO_2$–based ocean flux products to outline a new specification of ocean prior flux uncertainty based on the ensemble–spread of the different flux products (the "spread–based" uncertainty scheme). Towards the development of the spread–based scheme, we have compiled a set of eight global gridded interannually varying ocean–atmosphere $CO_2$ flux products. These are Landschutzer et al., 2016, Rodenbeck et al., 2014, Denvil–Sommer et al., 2019, Iida et al., 2015, Zeng et al., 2015, Gregor et al., 2019, Chau et al., 2020, and Watson et al., 2020.

The spread–based prior flux uncertainty scheme uses a diagnostic derived from the variation among the set of ocean atmosphere carbon flux products (see Eq. (1)). This scheme specifies lower uncertainty levels where alternative prior flux representations are in accord (e.g., when well–constrained by availability of surface $pCO_2$ measurements), and higher uncertainty levels where the prior flux distributions differ significantly (typically in under–sampled regions or those of significant flux variability). This specification follows previously used methods to characterize uncertainties in ocean flux distributions (e.g., Bopp et al., 2013). For this spread–based uncertainty specification, the gridded prior flux uncertainty, $U(i,j)$ (for a gridcell with coordinates $(i,j)$) is specified as the standard deviation of the spread of the different prior flux products. Thus, the uncertainty $U(i,j)$ is calculated as:

$$U(i,j) = sqrt(\sum_{k}^{K}\left(f_k(i,j) - \overline{f(\iota,j)}\right)^2)/(K-1) \tag{1}$$

Here K is the total number of the prior ocean flux products considered, and subscript k refers to an individual flux product. $f_k(i,j)$ represents the gridded monthly flux for each prior ocean flux and $\overline{f(\iota,j)}$ is the gridded monthly mean across all prior ocean flux products. These prior flux uncertainties are estimated on monthly timescales and also account for interannual variations. The uncertainty statistics of the prior ocean flux distributions will be dependent on the uncertainties associated with the respective inputs and methods of constructing the flux products. Ocean–atmosphere carbon flux products derived from surface ocean $pCO_2$ measurements are generally subject to two main sources of uncertainty: (i) in the specification of the surface $CO_2$ partial pressure difference across the air–sea interface, and (ii) in the specification of the gas–exchange coefficient used to derive fluxes (e.g., see discussion of Landschutzer et al., 2013; Watson et al., 2020). In the extended database of 8 $pCO_2$–based flux products that we present above, the majority of the flux products (seven of the eight) rely on the surface ocean $pCO_2$ data of the SOCAT database (Bakker et al., 2016). These flux products will be subject to similar uncertainties associated with data coverage in different ocean regions, although the uncertainties due to differences among surface interpolation methods may vary.

In this study we account for spatial correlations in the prior ocean fluxes, by inclusion of off diagonal elements in the background error covariance matrix $P^b$ (Appendix A Eq. A3). We follow the recommendations of Jones et al. (2012) on autocorrelation length scales in the surface ocean. That study derived spatial autocorrelation functions for air–sea fluxes from an analysis of the surface ocean $pCO_2$ database reported in Takahashi et al. (2009), combined with a gas exchange parameterization. We currently do not account for spatial correlation in land fluxes, but will investigate this in future analyses.

## 3 Results and Discussion

We first present in Sect. 3.1 results of short–term sensitivity tests that compare the influence of different prior ocean flux distributions and prior ocean flux uncertainty schemes on GCL estimates of North Atlantic (NA) $CO_2$ fluxes. Using these analyses as a basis, in Sect. 3.2 we conduct a multi–year GCL analysis of North Atlantic $CO_2$ ocean fluxes for the 2000–2017 period. We also report on derived characteristics of regionally aggregated North Atlantic subtropical and subpolar fluxes (long term means, trends and interannual variability) and compare these GCL results with recent estimates from other methodologies, including global ocean biogeochemical models (GOBMs), other atmospheric inverse studies, and surface $pCO_2$–based data products.

### 3.1 Sensitivity tests on specification of prior flux uncertainty

In this section we investigate, via sensitivity analyses, the application of the spread–based prior flux uncertainty scheme outlined in Sect. 2.4 in comparison to the fixed prior uncertainty levels commonly used in previous inverse estimates of ocean

CO$_2$ fluxes. The alternative specifications of prior flux uncertainty for ocean fluxes employed include (a) fixed percentage-based levels (U1:60% of prior flux, and U2:120% of prior flux), and (b) gridded flux uncertainties representing the variation or 'spread' of the different ocean flux data products at each location, and based on the standard deviation of the variation among the prior fluxes (U3: spread–based uncertainty; see Eq. (1)). The selection of the fixed percentage prior uncertainty levels used in the sensitivity analyses was based on the range of variability seen for the individual prior flux distributions (Fig. 1) for the sub–regions of the North Atlantic. These ranged from average levels of ~60% for the subtropical North Atlantic to levels greater than 120% for the subpolar North Atlantic, hence we have selected a level of U1:60% to characterize the lower sensitivity case, and U2:120% for the higher case. We apply the alternative flux uncertainty specifications to the three different ocean prior flux distributions discussed in Sect. 2.4, namely: (i) the Takahashi et al. (2009) climatology (Ta), (ii) the flux product of Landschützer et al. (2016) (La), and (iii) the flux product of Rödenbeck et al. (2014) (Ro).

Sensitivity analyses are conducted for the year 2003, following a 3 year GEOSChem model spin–up, starting from January 1$^{st}$, 2000; the length of spin–up was determined by recommendations on the duration required for stabilization of tropospheric CO$_2$ gradients (e.g., Gurney et al., 2002), and following methods used for previous GEOSChem CO$_2$ analyses (e.g., Nassar et al., 2010). The year 2003 was selected for sensitivity tests as the first viable year following spin–up. Analyses of inter–annual variability in Atlantic CO$_2$ (e.g., Landschutzer et al., 2013; Schuster et al., 2013) do not find 2003 to be an anomalous year for regional ocean fluxes. We evaluate the sensitivity of posterior ocean flux estimates with three different prior ocean uncertainty schemes U1, U2, and U3, described above; these are applied in turn for each of the three prior ocean flux distributions (Ta, La and Ro). Figure 1 presents the seasonal variation of the spatial distribution of the spread–based prior ocean flux uncertainty U3 (3 month averages for the year 2003). Figure 1 demonstrates that over the course of the year, and particularly in the Northern Hemisphere winter months, the spread–based uncertainty scheme (U3) provides a looser constraint on prior fluxes (i.e., levels of prior flux uncertainty > 120%) than the U1 and U2 schemes in the subpolar region, and a tighter constraint in the subtropical region (levels < 60%).

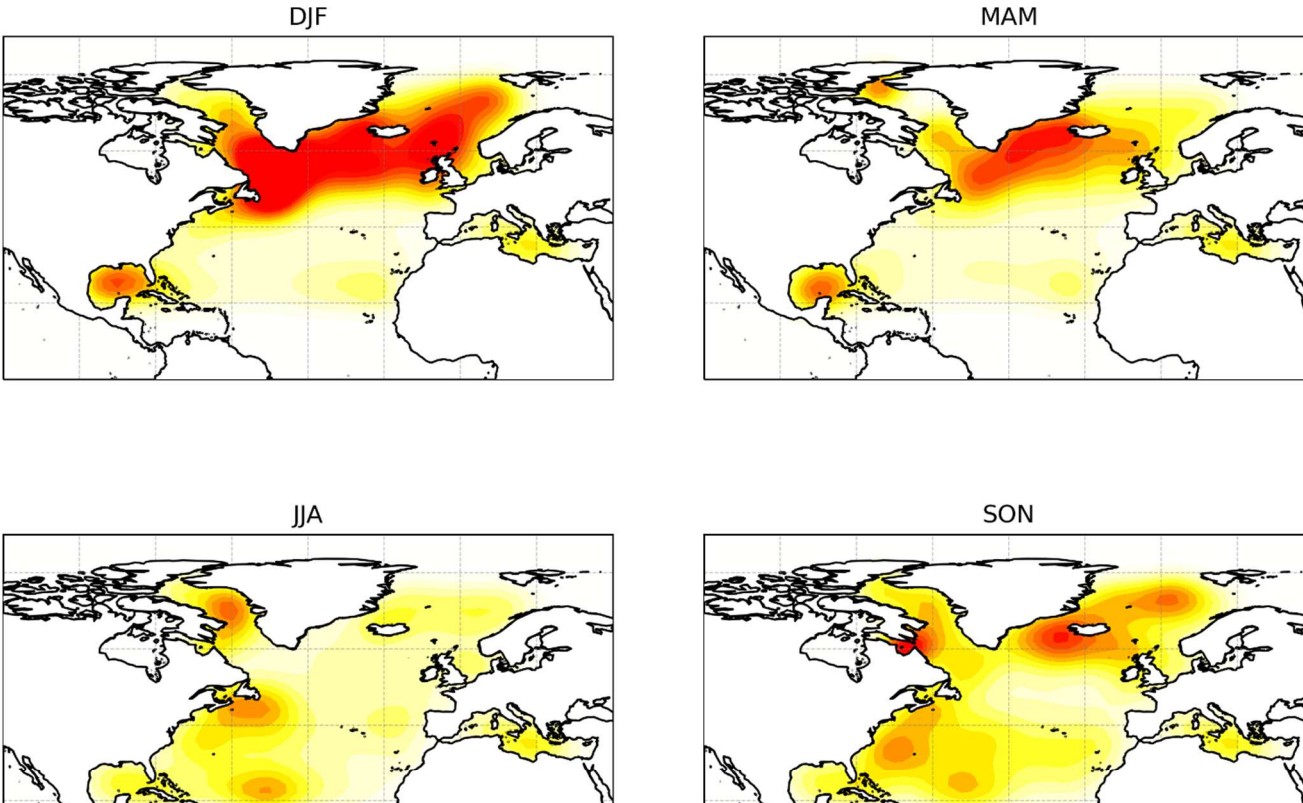

**Figure 1.** Distribution of the spread–based prior ocean flux uncertainty (U3) (3 month averages for the year 2003). The distribution in this study is calculated from the following 8 air–sea $CO_2$ flux products: (1) Denvil–Sommer et al., 2019 (product LSCE–FFNN–v1); (2) Iida et al., 2015 (JMA); (3) Zeng et al., 2015 (NIES); (4) Gregor et al., 2019 (CSIR–ML6); (5) Chau et al., 2020 (CMEMS); (6) Watson et al., 2020; (7) Landschützer et al., 2016; (8) Rödenbeck et al., 2014. It is represented here as a percentage of the prior ocean flux for ease of comparison with U1 and U2.  The percentage shown for each grid−cell is derived from the ratio of spread–based prior ocean uncertainty divided by the prior ocean flux value at that grid cell. DJF represents the monthly average for December, January, February; MAM for March, April, May; JJA for June, July, August; SON for September, October, November.

Table 1 summarizes the prior and posterior ocean flux estimates for the global and North Atlantic region (sub–divided into subpolar and subtropical regions) from the respective sensitivity tests. The distribution of prior flux for the subtropical North Atlantic shows closer agreement among the three source representations (Ta, La and Ro), with regional variation of 0.05 PgC $y^{-1}$, in comparison to a regional variation of ~0.1 PgC $y^{-1}$ for the subpolar region. Under the constraints provided by the atmospheric $CO_2$ observations all posterior flux estimates for the North Atlantic show increased uptake (Table 1), indicating

that all three representations of ocean prior flux underestimate the regional net atmosphere–ocean flux for the 2003 period. Largest changes in the regional posterior fluxes are estimated under the U3 specification of prior flux uncertainty. In addition, our estimates indicate a larger increase in $CO_2$ uptake in the subpolar basin (~0.05 PgC y$^{-1}$, changing from a prior flux range of -0.13 to -0.23 PgC y$^{-1}$ to posterior flux range of -0.18 to -0.27 PgC y$^{-1}$, for the U3 scenarios), in comparison to the smaller magnitude change for the subtropical North Atlantic basin (of ~ 0.04 PgC y$^{-1}$ from around -0.18 PgC y$^{-1}$ to -0.22 PgC y$^{-1}$ for the U3 scenarios)

**Table 1.** Global and North Atlantic $CO_2$ flux estimates from the GEOSChem−LETKF(GCL) system for year 2003 (PgC y$^{-1}$) summarizing sensitivity analyses on the prior ocean flux distribution and prior flux uncertainty. Prior flux references are Ta: Takahashi et al., 2009; La: Landschutzer et al., 2016; Ro: Rodenbeck et al., 2014. Prior flux uncertainty specifications are: U1: 60%; U2: 120%; U3: spread−based (following methods of Sect. 2.4).

| Global Ocean $CO_2$ Flux (PgC y$^{-1}$) | | | | | |
|---|---|---|---|---|---|
| Ta | -1.37 | La | -1.25 | Ro | -2.09 |
| TaU1 | -1.63±0.13 | LaU1 | -1.52±0.13 | RoU1 | -2.31±0.16 |
| TaU2 | -2.05±0.26 | LaU2 | -1.96±0.26 | RoU2 | -2.68±0.31 |
| TaU3 | -1.97±0.17 | LaU3 | -1.83±0.19 | RoU3 | -2.60±0.18 |
| North Atlantic subtropics [15°N−50°N] | | | | | |
| Ta | -0.22 | La | -0.18 | Ro | -0.17 |
| TaU1 | -0.23±0.02 | LaU1 | -0.19±0.02 | RoU1 | -0.18±0.02 |
| TaU2 | -0.25±0.05 | LaU2 | -0.21±0.04 | RoU2 | -0.20±0.04 |
| TaU3 | -0.26±0.03 | LaU3 | -0.22±0.03 | RoU3 | -0.23±0.03 |
| North Atlantic subpolar [50°N−80°N], eastern boundary at 20°E | | | | | |
| Ta | -0.23 | La | -0.13 | Ro | -0.21 |
| TaU1 | -0.23±0.05 | LaU1 | -0.13±0.02 | RoU1 | -0.22±0.04 |
| TaU2 | -0.25±0.1 | LaU2 | -0.14±0.05 | RoU2 | -0.23±0.09 |
| TaU3 | -0.27±0.05 | LaU3 | -0.18±0.05 | RoU3 | -0.24±0.05 |

We note that the increases in estimated uptake for the North Atlantic basins are relatively smaller (on average in the range 10–20%) than the increased uptake estimated on the global scale (~30–50% changes, see Table 1), indicating that the prior flux representations of North Atlantic carbon uptake are more consistent with the constraints from atmospheric $CO_2$ measurements than the comparison on a global scale.

The U3 flux uncertainty specification is derived from the variation among a set of ocean–atmosphere carbon flux products (Eq. (1)). This scheme specifies lower uncertainty levels where alternative prior flux representations are in accord (e.g., when well constrained by availability of surface $pCO_2$ measurements, as in the subtropical North Atlantic), and higher uncertainty levels where the prior flux distributions differ significantly (typically in under–sampled regions or those of significant flux variability, such as the subpolar North Atlantic). We further assess the value of the U3 scheme using a metric of GCL modeled

atmospheric $CO_2$ concentration; specifically, estimates of the model–observation mismatch for the year 2003 at the NOAA network station sites in the North Atlantic using the a posteriori fluxes associated with the sensitivity analyses of this section (Appendix Table A2). The results summarized in Table A2 indicates that scheme U3 provides the smallest magnitude model-observation mismatch for the individual North Atlantic sites and for the global network average. For the long term analyses of the remainder of this study, therefore, we use the U3 spread–based flux uncertainty scheme in preference to the fixed level flux uncertainty schemes used in many previous inverse analyses.

### 3.2 Multi–year analyses of North Atlantic $CO_2$ fluxes

In this section we present results of a multi–year GCL analysis (for the period 2000−2017), calculating regional estimates of North Atlantic $CO_2$ fluxes on annual to decadal timescales. Prior flux distributions for fossil emissions, and exchange with the land biosphere fluxes are as described in Sect. 2.4. For ocean prior fluxes, we employ the distribution of Landschützer et al. (2016); this is an established surface $pCO_2$−based product and also provides inter−annually varying fluxes over the entire estimation period (2000–2017), in comparison to the climatology−only fluxes of Takahashi et al. (2009). Ocean prior flux uncertainties are specified by the spread−based scheme U3 described above and derived from the eight ocean–atmosphere $pCO_2$–based flux products summarized in Sect. 2.4.

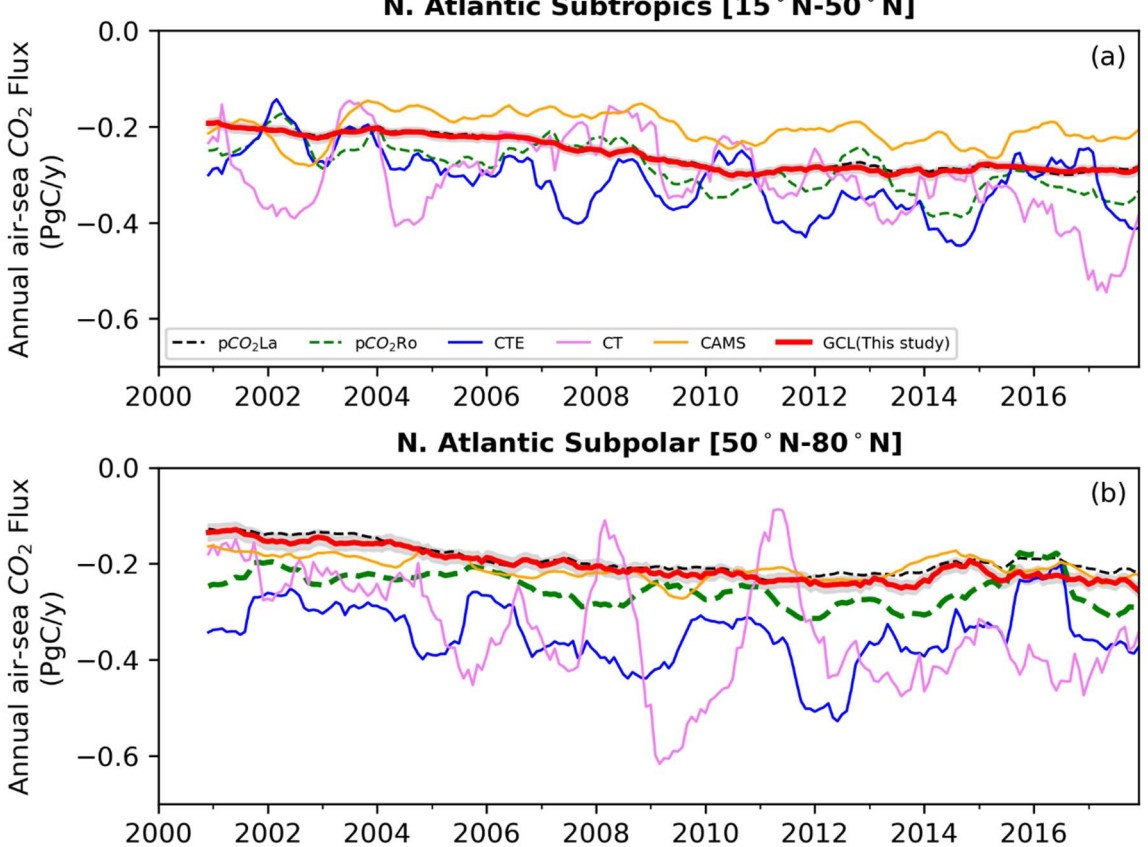

**Figure 2.** Comparison of annual air–sea $CO_2$ fluxes for North Atlantic for the 2000–2017 period for: (a) North Atlantic Subtropics; and (b) North Atlantic Subpolar regions. The GCL posterior flux estimate from this study (red) is derived from the prior flux of Landschützer et al., 2016 (pCO2La: black). The grey shaded area represents the uncertainty estimate on the GCL posterior flux (plotted at a 1 sigma level). Also shown are the flux estimates of (i) Chevallier et al., 2019 (CAMS: yellow); (ii) Jacobson et al., 2020 (CT: CarbonTracker2019: pink); and (iii) van der Laan–Luijkx et al., 2017 (CTE: Carbon Tracker Europe: blue). All time series shown have a 12 month running mean filter applied.

Figure 2 presents the variation of air–sea $CO_2$ flux for the North Atlantic subtropical and subpolar regions for the 2000–2017 period (represented as a 12 month running average). We also plot on Fig. 2 flux estimates from three other atmospheric inverse analysis studies including CAMS (v18r2, Chevallier et al., 2019), CT (CarbonTracker 2019, Jacobson et al., 2020 ) and CTE (Carbon Tracker Europe, van der Laan–Luijkx et al., 2017). All data are regridded to 2° latitude × 2.5° longitude to be consistent with the GCL model resolution.

For the North Atlantic subtropical region, the GCL posterior flux magnitude is close to that of the ocean prior flux employed (Landschutzer et al., 2016), with differences of approximately 0.01 PgC $y^{-1}$ over the period. Variation among the other inverse flux estimates can reach up to 0.3 PgC $y^{-1}$ (e.g., between CT and CAMS in 2017), and these differences can be ascribed, in part, to the different underlying prior flux distributions used in the respective inverse analyses (see Sect. 3.2.2). For the North

Atlantic subpolar region, the GCL posterior flux estimate deviates more from the prior flux estimate (e.g., showing differences of up to 0.04 PgC y$^{-1}$), especially for some years (2012–2017) of the analysis. The majority of flux estimates for the North Atlantic subpolar region are in closer accord (Fig. 2b) with differences of less than 0.2 PgC y$^{-1}$ (the CT estimate is an exception indicating variations of greater than 0.3 PgC y$^{-1}$ from the other estimates). A potential reason for the anomalous behaviour of the CT estimate in the North Atlantic is the underlying prior flux uncertainties used in the analysis which give a loose constraint on the prior ocean fluxes and allow the ocean fluxes deviate far from the prior influenced by the atmospheric $CO_2$ signals (Jacobson et al., 2020).

We also note that Peylin et al. (2013) have suggested that significant inter–annual variability in atmospheric inverse estimates is a potential indicator of 'flux leakage', where significant variability of terrestrial carbon fluxes in combination with sparse atmospheric sampling can result in misattribution of carbon flux estimates between land and ocean. To assess the significance of flux leakage in our GCL analyses, we have calculated estimates of the diagnostic recommended by Peylin et al. (2013) (i.e., the correlation between the annual total land and total ocean fluxes) for the Northern Hemisphere as a whole (Equator to 90ºN), and also by latitudinal region. Estimates of this diagnostic are relatively low for our GCL analyses (values of 0.2 and 0.5 for the subpolar and subtropical regions) indicating low potential for flux leakage. As a point of comparison, Peylin et al. (2013) note that six out of eleven atmospheric inverse analyses in their model inter–comparison reported correlation coefficients of greater than 0.5.

### 3.2.1 Long term mean

Figure 3 provides a comparison of the following GCL flux estimates and associated characteristics for the North Atlantic subtropical and subpolar regions for the period 2000–2017: (i) the long term mean of air–sea $CO_2$ flux estimates (the underlying data are tabulated in Table 2); (ii) the estimated inter–annual variability (IAV) of fluxes (Table 3); and (iii) the long term trends (Table 4). The IAV is calculated following methods of Rödenbeck et al. (2015) (i.e., derived from the standard deviation of the residuals of a 12 month running mean over the $CO_2$ flux time series).

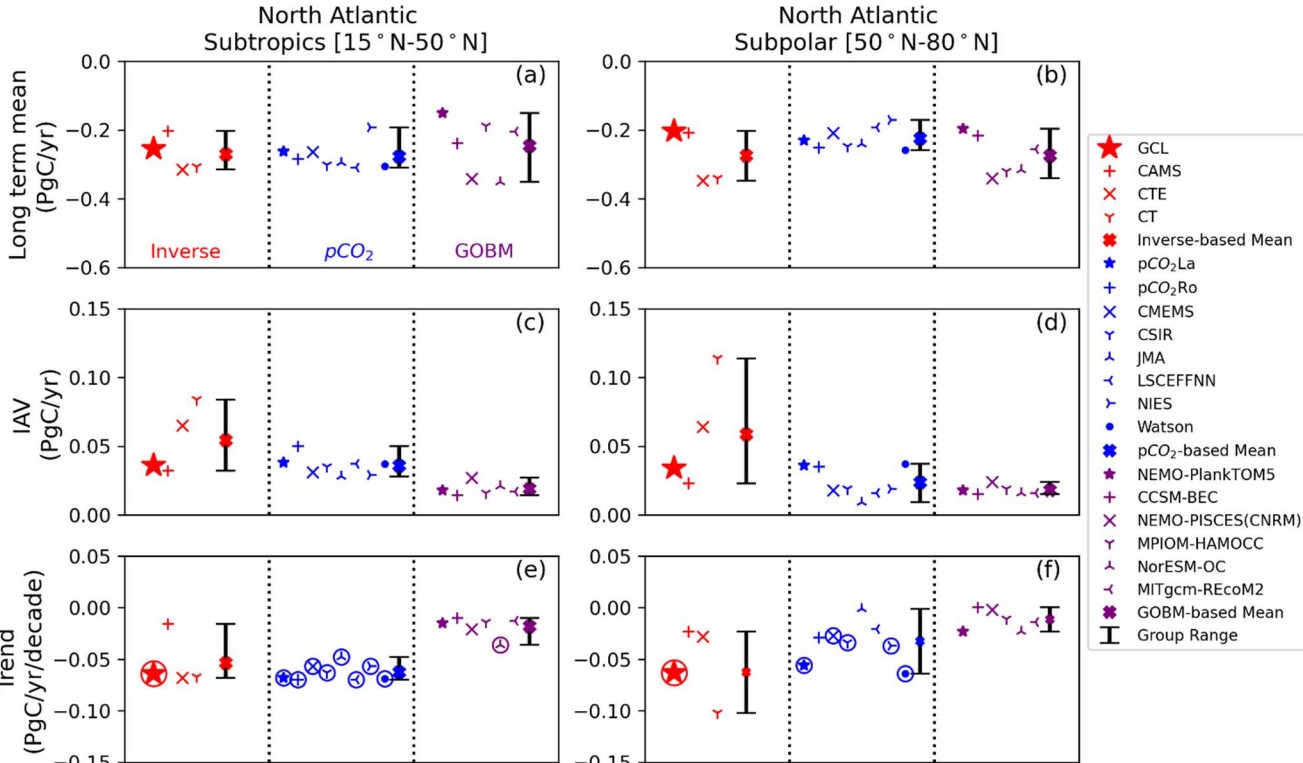

**Figure 3.** Comparison of $CO_2$ ocean flux metrics for the 2000−2017 period for North Atlantic subtropics (left panels) and subpolar regions (right panels). Metrics shown are the long term mean (panels (a) and (b)); interannual variability (IAV) (panels (c) and (d)); and long term trend (panels (e) and (f)). The GCL estimates (red stars) are shown in comparison to other atmospheric inverse analyses (red symbols), surface ocean $pCO_2$ products (blue) and global ocean biogeochemistry models (GOBMs, purple). Also shown are the estimated mean values from each sub–group of analyses (filled cross symbols) with their minimum–maximum range. Circled symbols in panel (e) and (f) indicate a statistically significant trend.

**Table 2.** Summary metrics of GEOSChem–LETKF North Atlantic (NA) $CO_2$ flux estimates, and comparison with independent estimates (from atmospheric inverse analyses, surface $pCO_2$ mappings, and Global Ocean Biogeochemistry models (GOBMs)) for the period 2000−2017. Listed are estimates for the long term mean. The metrics listed in this table are plotted in Fig.3a and 3b.

| Long term mean (PgC y$^{-1}$) | | |
|---|---|---|
| **NA Subtropics** (15°N–50°N) | **NA Subpolar** (50°N–80°N; eastern boundary at 20°E) | |
| **Atmospheric inversions** | | |
| *-0.255±0.037* | *-0.203±0.037* | *This study (GCL)*[a] |
| -0.203 | -0.208 | CAMS (Chevallier et al. 2019) |
| -0.315 | -0.347 | CTE (van der Laan–Luijkx et al. 2017) |
| -0.307 | -0.340 | CT (Jacobson et al. 2020) |
| [-0.315, -0.203] | [-0.347, -0.203] | Range of all atmospheric inverse studies (minimum to maximum) |
| **Surface ocean $pCO_2$–based flux products** | | |
| -0.263 | -0.230 | pCO$_2$La (Landschutzer et al. 2016) |
| -0.284 | -0.252 | pCO$_2$Ro (Rodenbeck et al. 2014) |
| -0.264 | -0.208 | CMEMS (Chau et al. 2020) |
| -0.302 | -0.248 | CSIR (Gregor et al. 2019) |
| -0.295 | -0.241 | JMA (Iida et al. 2015) |
| -0.309 | -0.192 | LSCEFFNN (Denvil–Sommer et al. 2019) |
| -0.193 | -0.171 | NIES (Zeng et al. 2015) |
| -0.305 | -0.259 | Watson et al. (2020) |
| [-0.309, -0.193] | [-0.259, -0.171] | Range of all pCO$_2$–based representations (minimum to maximum) |
| **Global ocean biogeochemistry models** | | |
| -0.150 | -0.197 | NEMO–PlankTOM5 (Buitenhuis et al. 2010) |
| -0.238 | -0.217 | CCSM–BEC (Doney et al. 2009) |
| -0.342 | -0.341 | NEMO–PISCES (CNRM) (Séférian et al. 2013) |
| -0.188 | -0.321 | MPIOM–HAMOCC (Ilyina et al. 2013) |
| -0.351 | -0.316 | NorESM–OC (Schwinger et al. 2016) |
| -0.205 | -0.256 | MITgcm–REcoM2 (Hauck et al. 2016) |
| [-0.351, -0.150] | [-0.341, -0.197] | Range of GOBM studies (minimum to maximum) |

[a] The uncertainty of the long term mean estimate from the GCL (this study) is calculated as the standard deviation of the annual flux estimates over the (2000–2017) period.

We also present in Fig. 3 the equivalent estimates from other independent assessments, including (i) other atmospheric inverse analyses, (ii) surface ocean $pCO_2$–based analyses, and (iii) analyses from global ocean biogeochemistry models (GOBMs). For the North Atlantic subtropical region, the long term mean of the GCL posterior flux estimate is -0.255±0.037 PgC $y^{-1}$ (Figure 3a and Table 2). It lies in the range spanned by the other inverse analyses (-0.31 to -0.20 PgC $y^{-1}$, of slightly larger magnitude than CAMS, but smaller than CT and CTE), and is in good agreement with the mean–value estimates from surface $pCO_2$–based methods and GOBMs. For the North Atlantic subpolar region, the GCL estimate of the long term mean uptake is -0.203±0.037 PgC $y^{-1}$ (Table 2), close to the inverse estimate of the CAMS analysis, and of smaller magnitude (by ~0.1 PgC $y^{-1}$) than the inverse estimates of CT and CTE. The GCL estimate is consistent with the mean estimate from $pCO_2$–based products, and within the range of flux estimates from GOBMs (from -0.341 to -0.197 PgC $y^{-1}$).

### 3.2.2 Interannual variability

The interannual variability (IAV) of $CO_2$ flux estimates derived from the GCL is 0.036±0.006 PgC $y^{-1}$ for the North Atlantic subtropics and 0.034±0.009 PgC $y^{-1}$ for the North Atlantic subpolar region (Fig. 3c and 3d, Table 3). The IAV estimates from the different inverse analyses for both the North Atlantic subtropics and subpolar regions display a larger range (0.032 to 0.084 PgC $y^{-1}$ and 0.023 to 0.114 PgC $y^{-1}$ respectively), than the ranges displayed by GOBMs (0.014 to 0.027 PgC $y^{-1}$ and 0.015 to 0.024 PgC $y^{-1}$ respectively) and $pCO_2$–based fluxes (0.029 to 0.050 PgC $y^{-1}$ and 0.009 to 0.037 PgC $y^{-1}$ respectively). The larger range of IAV from atmospheric inverse analyses is influenced especially by high magnitude IAV estimates from the CarbonTracker (CT) inverse analysis. Potential causes of the differences among atmospheric inversion between the GCL and CAMS IAV estimates and those of the CarbonTracker estimates are the different prior ocean fluxes employed by the inverse analyses, and the relative weighting assigned to the influence of atmospheric $CO_2$ observations (Jacobson et al., 2020). The GCL and CAMS estimates use the prior flux of Landschützer et al. (2016), CTE uses the prior flux of Rodenbeck et al. (2014) and the CarbonTracker inversions use the prior flux of Jacobson et al. (2007).

Recent synthesis studies of global ocean carbon fluxes have noted that GOBMs underestimate the magnitude of IAV in comparison to estimates from $pCO_2$–based mappings and inverse analyses (DeVries et al. 2019, Hauck et al. 2020). An important driver of IAV is the variability in biological carbon export; the lower variability observed in the GOBMs could result from opposing changes in biological vs. circulation impacts on carbon export, which potentially reduces the sensitivity of the GOBM air–sea carbon fluxes to climate variability (Landschutzer et al., 2013, DeVries et al., 2019).

**Table 3.** Summary metrics of GEOSChem–LETKF North Atlantic (NA) $CO_2$ flux estimates, and comparison with independent estimates (from atmospheric inverse analyses, surface $pCO_2$ mappings, and Global Ocean Biogeochemistry models (GOBMs)) for the period 2000–2017. Listed are estimates for the interannual variability of the regional fluxes over the period. The metrics listed in this table are plotted in Fig. 3 c, d.


| Interannual Variability (IAV) (PgC y⁻¹) | | |
|---|---|---|
| **NA Subtropics** (15°N–50°N) | **NA Subpolar** (50°N–80°N; eastern boundary at 20°E) | |
| **Atmospheric inversions** | | |
| *0.036±0.006* | *0.034±0.009* | *This study (GCL)[a]* |
| 0.032 | 0.023 | CAMS (Chevallier et al. 2019) |
| 0.065 | 0.064 | CTE (van der Laan–Luijkx et al. 2017) |
| 0.084 | 0.114 | CT (Jacobson et al. 2020) |
| [0.032, 0.084] | [0.023, 0.114] | Range of all atmospheric inverse studies (minimum to maximum) |
| **Surface ocean $pCO_2$−based flux products** | | |
| 0.038 | 0.036 | $pCO_2$La (Landschutzer et al. 2016) |
| 0.050 | 0.035 | $pCO_2$Ro (Rodenbeck et al. 2014) |
| 0.031 | 0.018 | CMEMS (Chau et al. 2020) |
| 0.035 | 0.019 | CSIR (Gregor et al. 2019) |
| 0.028 | 0.009 | JMA (Iida et al. 2015) |
| 0.037 | 0.016 | LSCEFFNN (Denvil–Sommer et al. 2019) |
| 0.029 | 0.019 | NIES (Zeng et al. 2015) |
| 0.037 | 0.037 | Watson et al. (2020) |
| [0.029, 0.050] | [0.009, 0.037] | Range of all $pCO_2$–based representations (minimum to maximum) |
| **Global ocean biogeochemistry models** | | |
| 0.018 | 0.018 | NEMO−PlankTOM5 (Buitenhuis et al. 2010) |
| 0.014 | 0.015 | CCSM-BEC (Doney et al. 2009) |
| 0.027 | 0.024 | NEMO-PISCES (CNRM) (Séférian et al. 2013) |
| 0.016 | 0.019 | MPIOM-HAMOCC (Ilyina et al. 2013) |
| 0.021 | 0.016 | NorESM-OC (Schwinger et al. 2016) |
| 0.017 | 0.016 | MITgcm-REcoM2 (Hauck et al. 2016) |
| [0.014, 0.027] | [0.015, 0.024] | Range of GOBM studies (minimum to maximum) |

[a] The uncertainty of the estimated IAV from the GCL (this study) is calculated as the standard deviation of the ensemble posterior fluxes

### 3.2.3 Estimated Trends of North Atlantic $CO_2$ Fluxes

Our calculations of estimated trends for the 2000–2017 period are presented in Table 4 and Fig. 3e and 3f. We also highlight in the table and figure panels, the trend estimates that are statistically significant (significant at the 95% confidence level, Montgomery et al., 2012). Our GCL analyses indicate statistically significant trends for the 2000–2017 period of -0.064±0.007 PgC $y^{-1}$ decade$^{-1}$ in the North Atlantic subtropical basin, and 0.063±0.008 PgC $y^{-1}$ decade$^{-1}$ in the subpolar region. These estimated trends are of similar magnitude to those estimated from surface ocean $pCO_2$ products, but of much larger magnitude

(by a factor of 3–4) than decadal trends estimated from the GOBMs (Fig. 3e, Table 4). Our findings are similar to those of Devries et al. (2019), who noted that decadal trend estimates of North Atlantic $CO_2$ uptake for the 2000s from the SOCOM inter–comparison of $pCO_2$–based flux products were larger than those from the GOBMS in their analysis (see Fig. 3 of their study).





**Table 4.** Summary metrics of GEOSChem–LETKF North Atlantic (NA) $CO_2$ flux estimates, and comparison with independent estimates (from atmospheric inverse analyses, surface $pCO_2$ mappings, and Global Ocean Biogeochemistry models (GOBMs)) for the period 2000–2017. Listed are estimates for the trend of the regional fluxes over the period. The metrics listed in this table are plotted in Fig. 3 e, f of the main study.

| Trend (PgC y$^{-1}$ decade$^{-1}$) | | |
| --- | --- | --- |
| **NA Subtropics** (15°N–50°N) | **NA Subpolar** (50°N–80°N; eastern boundary at 20°E) | |
| **Atmospheric inversions** | | |
| *-0.064±0.007 (S)[a]* | *-0.063±0.008(S)[a]* | *This study (GCL)[b]* |
| -0.016 | -0.023 | CAMS (Chevallier et al. 2019) |
| -0.068 | -0.028 | CTE (van der Laan–Luijkx et al. 2017) |
| -0.067 | -0.102 | CT (Jacobson et al. 2020) |
| [-0.016, -0.085] | [-0.023, -0.126] | Range of all Atmospheric inverse studies (minimum to maximum) |
| **Surface ocean pCO₂-based flux products** | | |
| -0.068(S) | -0.056(S) | pCO₂La (Landschutzer et al. 2016) |
| -0.070(S) | -0.029 | pCO₂Ro (Rodenbeck et al. 2014) |
| -0.057(S) | -0.027(S) | CMEMS (Chau et al. 2020) |
| -0.063(S) | -0.034(S) | CSIR (Gregor et al. 2019) |
| -0.048(S) | -0.001 | JMA (Iida et al. 2015) |
| -0.070(S) | -0.021 | LSCEFFNN (Denvil–Sommer et al. 2019) |
| -0.057(S) | -0.037(S) | NIES (Zeng et al. 2015) |
| -0.069(S) | -0.064(S) | Watson et al. (2020) |
| [-0.048, -0.070] | [-0.001,-0.064] | Range of all pCO₂–based representations (minimum to maximum) |
| **Global ocean biogeochemistry models** | | |
| -0.015 | -0.023 | NEMO–PlankTOM5 (Buitenhuis et al. 2010) |
| -0.010 | 0.0002 | CCSM–BEC (Doney et al. 2009) |
| -0.021 | -0.002 | NEMO–PISCES (CNRM) (Séférian et al. 2013) |
| -0.014 | -0.011 | MPIOM–HAMOCC (Ilyina et al. 2013) |
| -0.036(S) | -0.023 | NorESM–OC (Schwinger et al. 2016) |
| -0.013 | -0.014 | MITgcm–REcoM2 (Hauck et al. 2016) |
| [-0.010, -0.036] | [-0.0002, -0.023] | Range of GOBM studies (minimum to maximum) |

[a] The symbol (S) indicates that the calculated trend is statistically significant (at the 95% confidence interval).

[b] The uncertainty of the fitted trend from the GCL estimates is reported as 1 standard deviation of the OLS fitted slope (Montgomery et al. 2012).

## 4 Summary

In this study we present a new long term estimate of North Atlantic air–sea $CO_2$ fluxes for recent decades (period 2000–2017) using the atmospheric carbon cycle data assimilation system GEOSChem–LETKF. We focus, in particular, on the specification of prior ocean fluxes, including sensitivity of flux estimates to alternative prior flux distributions, and on the specification of uncertainties associated with ocean fluxes. Towards this we have developed the 'spread-based' flux uncertainty scheme which represents the variability among a set of different prior ocean $CO_2$ flux representations. The scheme ascribes higher levels of uncertainty to regions with larger discrepancies among ocean $CO_2$ prior flux representation that arise from uncertainties associated with measurement density and $pCO_2$–interpolation methods (Sect. 2.4). The spread-based flux uncertainty scheme provides improved performance in comparison to schemes with fixed prior flux uncertainty levels, based on an assessment metric of differences in model–observation values for atmospheric $CO_2$ at North Atlantic measurement sites of the NOAA–GLOBALVIEWCO$_2$ network (Sect. 3.1). It provides a valuable new means of specifying prior flux uncertainties for atmospheric inverse analyses of ocean $CO_2$ fluxes.

We have used the spread–based flux uncertainty scheme in the GEOSChem–LETKF to derive estimates of $CO_2$ fluxes in the North Atlantic for the 2000–2017 period. Long term mean estimates of the regional ocean $CO_2$ uptake are -0.255±0.037 PgC $y^{-1}$ for the North Atlantic subtropics and -0.203±0.037 PgC $y^{-1}$ for North Atlantic subpolar region, and are consistent with recent regional flux estimates from surface $pCO_2$–based methods and global ocean biogeochemistry models (GOBMs). The GEOSChem–LETKF estimates of interannual variability in air–sea $CO_2$ fluxes are 0.036±0.006 PgC $y^{-1}$ (North Atlantic subtropics) and 0.034±0.009 PgC $y^{-1}$ (North Atlantic subpolar). In common with estimates from other atmospheric $CO_2$ inverse studies, the magnitude of IAV derived from the GEOSChem–LETKF is larger than corresponding estimates from GOBMs. Our GEOSChem–LETKF estimates also indicate statistically significant trends of increasing $CO_2$ uptake for the North Atlantic subtropical and subpolar region (estimated trend of -0.064±0.007 and -0.063±0.008 PgC $y^{-1}$ decade$^{-1}$ respectively). These trends are of comparable magnitude to those estimated from surface $pCO_2$–based flux products, but much larger than those derived from global ocean biogeochemistry models for the 2000-2017 period. Estimates of inter–annual variability and long term trends derived from our GEOSChem–LETKF analyses are generally more robust for the North Atlantic subtropics than for the subpolar region, and characterized by smaller uncertainty bounds. Limiting factors affecting estimates for the North Atlantic subpolar region include higher levels of uncertainty associated with specification of prior fluxes (Fig. 1), and the observational uncertainty at the atmospheric measurement $CO_2$ sites in these high northern latitudes (Table A1). The number of regional atmospheric $CO_2$ measurement sites available to constrain North Atlantic subpolar fluxes are also relatively few in comparison to the subtropical region. Improved ocean $CO_2$ flux estimates and associated metrics for this North Atlantic region will be obtained by provision of additional high accuracy marine boundary layer $CO_2$ measurements for the region from fixed surface sites and from ships and buoys (Wanninkhof et al., 2019).

## Appendix A: The Local Ensemble Transform Kalman Filter (LETKF) system

Here we briefly describe the LETKF system used for estimation of surface $CO_2$ fluxes. The methodology follows that of Hunt et al. (2007) and Miyoshi et al. (2007), and additional detail is provided in these publications. The LETKF has been previously used in meteorological forecasting, and more recently in atmospheric $CO_2$ data assimilation (e.g., Liu et al. 2019, 2016; Kang et al. 2012). The LETKF provides iterative estimates of the time evolution of the system state, $x$, (here representing the gridscale surface carbon fluxes, of dimension $m$). Each step involves a forecast stage (based on a physical model of the system evolution) and a state estimation stage (the 'analysis' step), which combines system observations, $y$ (of dimension $n$), together with the background forecast, $x^b$, to derive the improved state estimate. The observation operator $H$ provides the mapping from the state space to the observation space; in this study $H$ is provided by the GEOSChem atmospheric model. In the analysis step, the surface carbon flux estimates are obtained by minimization of a cost function (Eq. A1) which accounts for deviations of the system state $x$, from the background forecast, $x^b$, and for the mismatch between observations ($y$) and their modeled representations ($Hx$):

$$J(x) = (x - x^b)^T P^{-1}(x - x^b) + (y - Hx)^T R^{-1}(y - Hx) \tag{A1}$$

$B$ represents the background flux covariance matrix, and $R$ represents the observation covariance matrix.

In the LETKF system, an ensemble of model simulations is used to calculate the sample mean and covariance of the system state; thus, the background state $x^b$ is given by $(x^{b(i)}: i = 1.2, \dots k)$ for k ensemble members. The sample mean $\bar{x}^b$ and covariance $P^b$ of the background state vector given by:

$$\bar{x}^b = k^{-1} \sum_{i=1}^{k} x^{b(i)} \tag{A2}$$

$$P^b = (k-1)^{-1} \sum_{i=1}^{k} (x^{b(i)} - \bar{x}^b)(x^{b(i)} - \bar{x}^b)^T \tag{A3}$$
$$= (k-1)^{-1} X^b (X^b)^T$$

$X^b$ is an $m \times k$ matrix whose $i$th column is $x^{b(i)} - \bar{x}^b$. $P^b$ is the background state covariance matrix ($m \times m$).

Similarly the analysis state is represented by the ensemble $(x^{a(i)}: i = 1,2, \dots k)$ with its sample mean and covariance given by:

$$\bar{x}^a = k^{-1} \sum_{i=1}^{k} x^{a(i)} \tag{A4}$$

$$P^a = (k-1)^{-1} \sum_{i=1}^{k} (x^{a(i)} - \bar{x}^a)(x^{a(i)} - \bar{x}^a)^T \tag{A5}$$
$$= (k-1)^{-1} X^a (X^a)^T$$

where $X^a$ is the $m \times k$ matrix whose ith column is $x^{a(i)} - \bar{x}^a$.

The analysis state and covariance $\bar{x}^a$ and $P^a$ are updated based on the background information $\bar{x}^b$ and observations $y$ through the following equations:

$$\bar{x}^a = \bar{x}^b + P^a H^T R^{-1}(y - Hx^b) \tag{A6}$$

$$P^a = (I + P^b H^T R^{-1} H)^{-1} P^b \tag{A7}$$

The ensemble $y^{b(i)}$ of background observation vectors is defined by:

$$y^{b(i)} = H(x^{b(i)}) \tag{A8}$$

$$H(\bar{x}^b + X^b w) \approx \bar{y}^b + Y^b w \tag{A9}$$

where $Y^b$ is the $n \times k$ matrix whose $i$th column is $(y^{b(i)} - \bar{y}^b)$, and $w$ is a Gaussian random vector with mean $\bar{w}^b = 0$ and covariance $\tilde{P}^b = (k-1)^{-1} I$ .Then the analogues of analysis equations (6) and (7) are:

$$\bar{w}^a = \tilde{P}^a (Y^b)^T R^{-1}(y - \bar{y}^b) \tag{A10}$$

$$\tilde{P}^a = [(k-1)I + (Y^b)^T R^{-1} Y^b]^{-1} \tag{A11}$$

Following Hunt et al. (2007) and Miyoshi et al. (2007) (refer to these publications for the complete LETKF derivation) the overall analysis equation is:

$$x = \bar{x}^b + X^b[\tilde{P}^a (Y^b)^T R^{-1}(y - \bar{y}^b) + [(k-1)\tilde{P}^a]^{1/2}] \tag{A12}$$

The LETKF allows flexibility in the choice of observations to be assimilated at each grid point, based on the distance ($r$) of the observations from the gridpoint. The localization weighting function $f(r)$ is given by:

$$f(r) = \exp\left(-\frac{r^2}{2L^2}\right) \tag{A13}$$

where $L$ is an observation localization length which can be predefined to determine the outer boundary of the influence of the observations; i.e., the localization weighting function drops to zero at a value of

$$r = 2.\sqrt{\frac{10}{3}} L \tag{A14}$$

The observation localization is realized by multiplying the inverse of the localization function $f(r)$ with the observational error covariance $R$.

The parameter L represents the horizontal localization radius, and is set to 2000 km for this study, following Liu et al. (2016). The localization radius is used in the LETKF in a latitude–dependent weighting function which characterizes the spatial scale of the region within which atmospheric $CO_2$ observations are assimilated at each gridpoint (Miyoshi et al. 2007).

**Table A1.** Atmospheric $CO_2$ measurement sites[a]

| Site code | Longitude (degrees) | Latitude (degrees) | Altitude (m) | Site name | U[b] (ppm) |
|---|---|---|---|---|---|
| ABP | -38.16 | -12.76 | 6 | Arembepe, Bahia | 1.04 |
| ALT | -62.51 | 82.45 | 195 | Alert, Nunavut | 1.34 |
| AMY | 126.33 | 36.54 | 125 | Anmyeon-do | 8.88 |
| ASC | -14.40 | -7.97 | 90 | Ascension Island | 0.66 |
| ASK | 5.63 | 23.26 | 2715 | Assekrem | 0.80 |
| AZR | -27.08 | 38.75 | 24 | Terceira Island, Azores | 2.26 |
| BAL | 16.67 | 55.50 | 28 | Baltic Sea | 5.50 |
| BCS | -110.20 | 23.30 | 14 | Baja California Sur | 3.42 |
| BGU | 3.23 | 41.97 | 13 | Begur | 3.93 |
| BHD | 174.87 | -41.41 | 90 | Baring Head Station | 1.12 |
| BKT | 100.32 | -0.20 | 875 | Bukit Kototabang | 3.49 |
| BME | -64.65 | 32.37 | 17 | St. Davids Head, Bermuda | 2.57 |
| BMW | -64.88 | 32.27 | 60 | Tudor Hill, Bermuda | 2.12 |
| BRW | -156.60 | 71.32 | 28 | Barrow Atmospheric Baseline Observatory | 1.88 |
| BSC | 28.67 | 44.18 | 5 | Black Sea, Constanta | 9.88 |
| CBA | -162.72 | 55.20 | 25 | Cold Bay, Alaska | 2.41 |
| CFA | 147.06 | -19.28 | 5 | Cape Ferguson, Queensland | 1.04 |
| CGO | 144.68 | -40.68 | 164 | Cape Grim, Tasmania | 0.40 |
| CHR | -157.15 | 1.70 | 5 | Christmas Island | 0.60 |
| CIB | -4.93 | 41.81 | 850 | Centro de Investigacion de la Baja  Atmosfera (CIBA) | 3.97 |
| CPT | 18.49 | -34.35 | 260 | Cape Point | 0.74 |
| CRI | 73.83 | 15.08 | 66 | Cape Rama | 3.47 |
| CRZ | 51.85 | -46.43 | 202 | Crozet Island | 0.49 |
| CYA | 110.52 | -66.28 | 55 | Casey, Antarctica | 0.29 |
| DRP | -64.91 | -55.00 | 10 | Drake Passage | 0.41 |
| DSI | 116.73 | 20.70 | 8 | Dongsha Island | 3.46 |
| EIC | -109.45 | -27.15 | 55 | Easter Island | 1.80 |
| ELL | 0.96 | 42.58 | 2005 | Estany Llong | 2.41 |
| ESP | -126.53 | 49.38 | 47 | Estevan Point, British Columbia | 1.49 |
| FKL | 25.67 | 35.34 | 150 | Finokalia, Crete | 3.34 |

| | | | | | |
|---|---|---|---|---|---|
| GMI | 144.66 | 13.39 | 6 | Mariana Islands | 2.22 |
| GPA | 131.05 | -12.25 | 37 | Gunn Point | 2.02 |
| HBA | -26.21 | -75.61 | 35 | Halley Station, Antarctica | 0.16 |
| HPB | 11.02 | 47.80 | 990 | Hohenpeissenberg | 6.71 |
| HSU | -124.73 | 41.05 | 8 | Humboldt State University | 5.78 |
| HUN | 16.65 | 46.95 | 344 | Hegyhatsal | 6.00 |
| ICE | -20.29 | 63.40 | 127 | Storhofdi, Vestmannaeyjar | 2.03 |
| IZO | -16.48 | 28.30 | 2378 | Izana, Tenerife, Canary Islands | 1.21 |
| KEY | -80.20 | 25.67 | 6 | Key Biscayne, Florida | 4.14 |
| KUM | -154.82 | 19.52 | 8 | Cape Kumukahi, Hawaii | 1.77 |
| KZD | 75.57 | 44.45 | 412 | Sary Taukum | 3.19 |
| KZM | 77.88 | 43.25 | 2524 | Plateau Assy | 3.00 |
| LJO | -117.26 | 32.87 | 20 | La  Jolla, California | 2.72 |
| LLB | -112.45 | 54.95 | 546 | Lac La Biche, Alberta | 8.91 |
| LLN | 120.86 | 23.46 | 2867 | Lulin | 5.27 |
| LMP | 12.61 | 35.51 | 50 | Lampedusa | 2.08 |
| MAA | 62.87 | -67.62 | 42 | Mawson Station, Antarctica | 0.32 |
| MEX | -97.31 | 18.98 | 4469 | High Altitude Global Climate Observation Center | 1.33 |
| MHD | -9.90 | 53.33 | 26 | Mace Head, County Galway | 3.23 |
| MID | -177.37 | 28.22 | 8 | Sand Island, Midway | 1.39 |
| MKN | 37.30 | -0.06 | 3649 | Mt. Kenya | 1.98 |
| MLO | -155.58 | 19.53 | 3402 | Mauna Loa, Hawaii | 0.63 |
| MQA | 158.97 | -54.48 | 13 | Macquarie Island | 0.33 |
| NAT | -35.26 | -5.52 | 20 | Farol De Mae Luiza Lighthouse | 1.44 |
| NMB | 15.03 | -23.58 | 461 | Gobabeb | 1.13 |
| NWR | -105.58 | 40.05 | 3526 | Niwot Ridge, Colorado | 1.88 |
| OBN | 36.60 | 55.12 | 484 | Obninsk | 6.49 |
| OTA | 142.82 | -38.52 | 50 | Otway, Victoria | 17.45 |
| OXK | 11.81 | 50.03 | 1185 | Ochsenkopf | 8.18 |
| PAL | 24.12 | 67.97 | 570 | Pallas-Sammaltunturi, GAW Station | 3.72 |
| PDM | 0.14 | 42.94 | 2877 | Pic Du Midi | 2.71 |
| POC | -145.13 | 14.97 | 20 | Pacific Ocean | 1.47 |
| PSA | -64.00 | -64.92 | 15 | Palmer Station, Antarctica | 0.23 |

| | | | | | |
|---|---|---|---|---|---|
| PTA | -123.73 | 38.95 | 22 | Point Arena, California | 5.50 |
| RK1 | -177.90 | -29.20 | 12 | Kermadec Island | 2.23 |
| RPB | -59.43 | 13.17 | 20 | Ragged Point | 0.83 |
| SDZ | 117.12 | 40.65 | 298 | Shangdianzi | 9.57 |
| SEY | 55.53 | -4.68 | 7 | Mahe Island, Seychelles | 0.98 |
| SGP | -97.48 | 36.62 | 374 | Southern Great Plains, Oklahoma | 4.91 |
| SHM | 174.10 | 52.72 | 28 | Shemya Island, Alaska | 2.91 |
| SIS | -1.26 | 60.09 | 33 | Shetland Islands | 2.87 |
| SMO | -170.57 | -14.25 | 47 | Tutuila, American Samoa | 1.19 |
| STM | 2.00 | 66.00 | 7 | Ocean Station M | 2.03 |
| SUM | -38.42 | 72.60 | 3215 | Summit | 1.32 |
| SYO | 39.58 | -69.00 | 16 | Syowa Station, Antarctica | 0.23 |
| TAC | 1.14 | 52.52 | 236 | Tacolneston | 6.78 |
| TAP | 126.13 | 36.73 | 21 | Tae-ahn Peninsula | 6.90 |
| THD | -124.15 | 41.05 | 112 | Trinidad Head, California | 4.54 |
| TIK | 128.89 | 71.60 | 29 | Hydrometeorological Observatory of Tiksi | 2.64 |
| USH | -68.31 | -54.85 | 32 | Ushuaia | 1.41 |
| UTA | -113.72 | 39.90 | 1332 | Wendover, Utah | 2.65 |
| UUM | 111.10 | 44.45 | 1012 | Ulaan Uul | 2.78 |
| WIS | 34.78 | 30.86 | 482 | Weizmann Institute of Science at the Arava Institute, Ketura | 2.39 |
| WLG | 100.92 | 36.27 | 3815 | Mt. Waliguan | 2.26 |
| WPC | 167.50 | -29.86 | 10 | Western Pacific Cruise | 1.70 |
| ZEP | 11.89 | 78.91 | 479 | Ny-Alesund, Svalbard | 1.82 |


[a] Source reference: Cooperative Global Atmospheric Data Integration Project, 2018. Version: obspack_co2_1_GLOBALVIEWplus_v4.2_2019-03-19 (https://doi.org/10.25925/20190319 )

[b] The specification of observational uncertainty U on atmospheric $CO_2$ measurements (and represented in matrix $R$ of Eq. A1) is calculated

as the standard deviation of measurement variability and using the detrended and deseasonalized $CO_2$ time series at each measurement site (following methods of Chevallier et al., 2010).

**Table A2**: Model–Observation mismatch in atmospheric $CO_2$ concentrations (unit: ppm) at North Atlantic sites (average over year 2003). GCL model values are derived from the a posteriori model analyses associated with the sensitivity analyses of Sect. 3.1. Atmospheric $CO_2$ observations are from the NOAA–GLOBALVIEW network described in Sect. 2.3.

| Sensitivity Analyses (Sect. 2.3) | North Atlantic Sites | | | | | | Global Network Average |
|---|---|---|---|---|---|---|---|
| | BMW | KEY | AVI | AZR | IZO | ICE | |
| U1Ta | 0.81 | 0.88 | 0.76 | 1.49 | 1.83 | 1.20 | 0.54 |
| U2Ta | 0.71 | 0.82 | 0.69 | 1.42 | 1.74 | 1.13 | 0.44 |
| U3Ta | 0.58 | -0.20 | 0.36 | 0.85 | 1.64 | 0.40 | 0.01 |
| U1La | 0.92 | 0.93 | 0.76 | 1.60 | 1.94 | 1.44 | 0.61 |
| U2La | 0.84 | 0.87 | 0.68 | 1.53 | 1.86 | 1.38 | 0.51 |
| U3La | 0.52 | -0.27 | 0.21 | 0.84 | 1.58 | 0.58 | 0.04 |
| U1Ro | 0.74 | 0.74 | 0.54 | 1.35 | 1.66 | 0.98 | 0.35 |
| U2Ro | 0.66 | 0.68 | 0.47 | 1.28 | 1.57 | 0.91 | 0.26 |
| U3Ro | 0.55 | -0.28 | 0.21 | 0.73 | 1.48 | 0.27 | -0.11 |

*Data Availability.*

Data sources: (i) Atmospheric $CO_2$ measurements were taken from obspack_co2_1_GLOBALVIEWplus_v4.2_2019-03-19 (https://doi.org/10.25925/20190319 ); (ii) Prior ocean flux oc_v1.7 from Rödenbeck et al . (2013) taken from http://www.bgc-jena.mpg.de/CarboScope/. Prior ocean flux Landschützer et al. (2016) taken from https://www.nodc.noaa.gov/ocads/oceans/SPCO2_1982_present_ETH_SOM_FFN.html. Prior ocean flux from Takahashi et

al. (2009) taken from ftp://ftp.as.harvard.edu/gcgrid/geos-chem. (iii) CarbonTracker CT2019 results provided by NOAA ESRL, Boulder, Colorado, USA from the website at http://carbontracker.noaa.gov. CTE flux estimates downloaded from ftp://ftp.wur.nl/carbontracker/data/fluxes/data_flux1x1_monthly/ on 24 November 2020. The flux estimates from CAMS (v18r2) taken from https://apps.ecmwf.int/datasets/data/cams-ghg-inversions/. (iv) The model $CO_2$ fluxes for JULES (land) and GOBMs (ocean) taken from (Le Quéré et al., 2018). Time series of reconstructed surface ocean $p$CO$_2$ and $CO_2$ fluxes

(LSCEFFNN) from Denvil–Sommer et al., 2019 are the first version of CMEMS, downloaded from http://marine.copernicus.eu/services-portfolio/access-to-products/. The products from Iida et al., 2015 downloaded from http://www.data.jma.go.jp/gmd/kaiyou/english/co2_flux/co2_flux_data_en.html. The products from Zeng et al., 2015 downloaded from https://db.cger.nies.go.jp/DL/10.17595/20201020.001.html.en. The products from CMEMS, CSIR, Watson taken from Friedlingstein et al., 2020.

*Author contributions.* ZC and PS designed the study. ZC, PS, JZ and NZ developed the model. ZC, PS, AW, and US discussed the design of simulations. ZC performed the simulations and analysis and wrote the initial manuscript. All authors contributed to the writing of the paper.

*Competing interests.* The authors declare that they have no conflict of interest.

*Disclaimer.* The work reflects only the author's view,the Europen Commission and their executive agency are not responsible for any use that may be made of the information the work contains.

*Acknowledgements.* This work was performed using the High Performance Computing Cluster at the University of East Anglia.

*Financial support.* It was supported through the UK Natural Environment Research Council grants NE/K002473/1(RAGNARoCC).

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
