# Peer review of "Variability of North Atlantic CO2 fluxes for the 2000–2017 period estimated from atmospheric inverse analyses"

_Biogeosciences, 2020_

## Referee Comment (RC1) · Anonymous Referee #1 · 3 Dec 2020

Summary

The authors present an improved atmospheric inversion data assimilation model (GCL) and apply it to the investigation of mean, variability, and trends of North Atlantic air-sea carbon fluxes. Specifically, the advancements made to the inverse model within involve multiple representations of prior ocean fluxes as well as sensitivity experiments assessing the different priors and related flux uncertainties from three different schemes. Additionally, comparisons are made to previous estimates of North Atlantic carbon fluxes as well as estimates from observation-based pCO2 products and global ocean models. Overall, I found this manuscript well organized, concise, and novel. I would support its publication but have a few suggestions that I believe that would improve the overall strength of the paper.

[Figure]

Main comments

I highly suggest including additional observation-based products in your analysis (Figure 3 specifically). You use Takahashi et al 2009 in the uncertainty analysis section but cant utilize it for long term mean/variability because it is of course only a climatology. You include 2 products (one by Landschützer and one by Rödenbeck) but there are more available and I highly suggest including them in the comparison to improve your message. Given that your ensemble of inverse models and of GOBMs are much larger than 2, it is worth making the effort to include more pCO2 products as well. See Denvil-Sommer et al. 2019, Gregor et al. 2019, Iida et al. 2015, and Zeng et al. 2015 for starters. Additionally, if Landschützer's product is used as the prior for the GCL inverse method, is it fair to use it as an independent comparison? If the data-assimilation method is trying to "fit" or "correct" the GCL inverse model to the pCO2 from that product then I would not consider it an independent comparison.

Section 2.5 could use more discussion/explanation. To my understanding, while the inverse model itself is not new, this method of specifying prior CO2 fluxes and using them to create more robust flux uncertainties seems to be a major improvement described in this manuscript. I'd be keen to see more explanation and discussion on that in this section.

Lastly, I think the title could be more descriptive of the actual work you are presenting. Specifically mentioning inverse models or uncertainty or a comparison between approaches.

Minor comments

The end of the introduction could improve from a motivation statement. Why do this work? Who will use this? How will it impact the community and what are the broader impacts? Clearly your improvements on the uncertainty estimates would be beneficial to the community as a whole so make that case more clearly.

[Figure]
Why is year 2003 selected for sensitivity tests on the prior flux uncertainty? Is three years of spin up sufficient? Is 2003 an anomalous year at all? With the dynamics at play in the North Atlantic basin it is important to consider how the selection of one year of focus can influence your analysis.

It could be clarified that when you move on from Section 3.1 you will only be using the U3 approach to specify uncertainty. Additionally, the same for your selection of Landschützer et al. 2017 as the prior for the GCL model as you transition to analysis in Section 3.2.

Figure 2: I find it very interesting that the CTE is so anomalous in the NA subtropics but the CT model is more anomalous in the subpolar regions. It jumps out at you from this figure and you barely notice anything else. Would be worth further discussion as to why those are so different in their mean, IAV, and decadal variability. What do these other inverse methods use as prior flux inputs?

Figure 3: could be cleaned up and simplified by reducing the y-label axis ticks and tick labels. Additionally, on this figure, if the trends in subplots e and f are not significant, consider making the filling color gray or something else to distinguish. They should be noted on the figure as well as in the table. Currently you only note that one of the GCL trends is significant and one is not but need to do this for all inverse models, pCO2 products, and GOBMs as well.

Throughout the paper, where you mention Figure 3, please add the subplot letter so that the reader can easily navigate to which subplot you are referencing (e.g. Line 268: "Fig 3e, Table 4)". Also, in the Table 2 caption you could reference "Figure 3 a,b" rather than just "Fig 3".

Section 3.2.2. should include further discussion and references explaining why the GOBMs have such low IAV as compared to the other products and inverse models.

Line 225: Your first comparison is to Schuster et al. 2013 but that work is looking at a

very different time period. While it can still be referenced and mentioned, highlighting comparisons that focus on the same decades of analysis is more appropriate.

Your summary statement on beginning on Line 303 could be expanded on. How is it more "robust"? Is it just smaller uncertainties and significant trends? Perhaps tie in reference to Figure 1 to discuss further.

References

Denvil-Sommer, A., Gehlen, M., Vrac, M., and Mejia, C.: LSCEFFNN-v1: a two-step neural network model for the reconstruction of surface ocean pCO2 over the global ocean, Geosci. Model Dev., 12, 2091–2105, https://doi.org/10.5194/gmd-12- 2091-2019, 2019.

Gregor, L. et al. Geosci. Model Dev., 12, 5113–5136, 2019 https://doi.org/10.5194/gmd-12-5113-2019

Y. Iida, A. Kojima, Y. Takatani, T. Nakano, T. Midorikawa, and M. Ishii: Trends in pCO2 and sea-air CO2 flux over the global open oceans for the last two decades. Journal of Oceanography, doi:10.1007/s10872-015-0306-4 (2015).

Zeng, J., Nojiri, Y., Nakaoka, S.‐i., Nakajima, H. and Shirai, T. (2015), Surface ocean CO2 in 1990–2011 modelled using a feed‐forward neural network. Geosci. Data J., 2: 47-51. https://doi.org/10.1002/gdj3.26

---

## Referee Comment (RC2) · Anonymous Referee #2 · 18 Jan 2021

The authors have studied the CO$_2$ annual fluxes in the North Atlantic during an 18-yr period with an atmospheric inverse modelling approach. They show some agreement with other estimates and present a sensitivity study with respect to the prior ocean flux constraint. The topic is obviously of great interest but the actual paper is rather deceiving, with little scientific depth. I am listing here important questions that are fully in the paper scope but that seem to be left open:

- How significant are the presented sensitivity tests for the inversion community? Despite a subsection and an appendix devoted to it, the description of the data assimilation system is unclear on what matters in practice. My interpretation of l. 95 is that the elementary assimilation window of the LETKF is of four weeks, a period which is too short (given mixing time scales in the atmosphere) to allow a clear distinction between the uncertainty in the prior initial state of atmospheric $CO_2$ and the uncertainty in the prior surface fluxes, when assimilating atmospheric measurements. The authors should therefore not separate the two. However, Incidentally, in the legend of Eq. A1 in the appendix, we understand that the uncertainty in the initial state of atmospheric $CO_2$ has been neglected. This rough simplification makes it hard to interpret B, officially the flux covariance matrix, in these terms.

- Similarly, the authors do not discuss spatial correlations in the prior errors, leaving the impression that they have neglected them as well. How credible is this hypothesis, e.g., among the prior ocean flux products tested here?

- How are the ocean flux results presented here affected by the leakage from the land fluxes noted in l. 48? The statement in l. 156 suggests there is none of significance, but without any justification.

Additionally, a number of points of various important need clarification:

- L. 27: the 20% value is rather artificial given the fact that the global ocean uptake is made of both sources and sinks.

- L. 57-8: bad example; the studies mentioned here are not for the same year and therefore should not use the same uncertainty budget for a frozen prior flux distribution anyway, given existing trends in the real fluxes.

- L. 91: this is Appendix A, not A1.

- L. 121: what is the rationale behind the 60% and 120% values? The authors should relate them to their knowledge of the quality of their prior fluxes, while they make it look arbitrary (except if indeed matrix B is just an ensemble of tuning factors and not an error covariance matrix; see above).

- L. 127: what is the value of K? I get the impression that only 3 flux products are used here: no standard deviation can be estimated from just three members.

- L. 140-1: why would the three prior ocean flux distributions have the same uncertainty statistics?

- L. 170: flexibility is not the question. The question is about well modelling the prior uncertainty.

- Table 2: if the numbers behind plus/minus signs for the mean values across studies are standard deviations, how can they have been computed on 6, 3, or even 2 members only?

- L. 339: what is the value of L?

---

## Author Comment (AC1) · 22 Feb 2021

We thank the reviewer for your assessment of our paper manuscript and the useful comments to improve the text. We have uploaded a pdf file as supplement that provides a response to each of the comments suggested by the reviewer.

Please also note the supplement to this comment:
https://bg.copernicus.org/preprints/bg-2020-385/bg-2020-385-AC1-supplement.pdf

---

## Author Comment (AC2) · 22 Feb 2021

Author responses to comments of the 2 referees.

Reviewer comments are listed in italics.

We thank both the reviewers for their detailed comments and suggestions on improvements to the manuscript. Below we list our responses to both sets of reviewer comments.

**REVIEWER 1**
*The authors present an improved atmospheric inversion data assimilation model (GCL) and apply it to the investigation of mean, variability, and trends of North Atlantic air-sea carbon fluxes. Specifically, the advancements made to the inverse model within involve multiple representations of prior ocean fluxes as well as sensitivity experiments assessing the different priors and related flux uncertainties from three different schemes. Additionally, comparisons are made to previous estimates of North Atlantic carbon fluxes as well as estimates from observation-based pCO₂ products and global ocean models. Overall, I found this manuscript well organized, concise, and novel. I would support its publication but have a few suggestions that I believe that would improve the overall strength of the paper.*

*Reviewer 1: Main comments*
*I highly suggest including additional observation-based products in your analysis (Figure 3 specifically). You use Takahashi et al 2009 in the uncertainty analysis section but can't utilize it for long term mean/variability because it is of course only a climatology. You include 2 products (one by Landschützer and one by Rödenbeck) but there are more available and I highly suggest including them in the comparison to improve your message. Given that your ensemble of inverse models and of GOBMs are much larger than 2, it is worth making the effort to include more pCO₂ products as well. See Denvil-Sommer et al. 2019, Gregor et al. 2019, Iida et al. 2015, and Zeng et al. 2015 for starters.*

**Response:** We agree with the reviewer's suggestion, that inclusion of additional observation-based products would strengthen our analysis. Accordingly we have sourced the following additional global pCO₂-based air-sea CO₂ flux products, and include them in our analyses. The specific data product names, or model versions are appended within parentheses following each published source.
(1) Denvil-Sommer et al. 2019 (product LSCE-FFNN-v1); (2) Iida et al. 2015 (JMA); (3) Zeng et al. 2015 (NIES); (4) Gregor et al. 2019 (CSIR-ML6); (5) Chau et al. 2020 (CMEMS); and (6) Watson et al. 2020.

The above six data products all account for inter-annually varying air-sea CO₂ fluxes (as do our previously included data products from Landschützer et al. (2016) and Rödenbeck et al. (2013)). Some initial updates of our results, accounting for the inclusion of these six additional flux products, are presented in the response below and in subsequent sections of our Response to Reviewers. Updated results included in this Response to Reviewers include Figure 1, Figure 3 and revised sections of Tables 2, 3, and 4, that now include the additional pCO₂-based flux products. Due to the significant additional computational resources required, we have not, as yet, revised the multi-year (2000-2017) GEOSChem-LETKF analysis reported in section 3.2, to include the updated 8-flux spread-based uncertainty shown in the revised Figure 1B. We can update this multi-year analysis if required.

Updated results for Figure 1 of our study: We present below the original version of Figure 1 (three flux products), along with a revised version (eight flux products).

- Figure 1A: Original version of Figure 1 for the three flux products from Landschützer et al. 2016, Rödenbeck et al. 2013, and Takahashi et al. 2009.
- Figure 1B : Spread-based prior ocean flux uncertainty for an extended set of eight flux products that only include the data products that represent inter-annually varying fluxes (i.e., the climatological product of Takahashi et al. 2009, is not included in Figure 1B).

We note that the main features and magnitudes of the spread-based prior flux uncertainty are consistent across the two figures, for example, highest levels of flux uncertainty are associated with the sub-polar North Atlantic region for the winter (DJF) and spring (MAM) months. This similarity is not unexpected given the common underlying $pCO_2$ database (i.e., SOCAT; Bakker et al., 2016, 2020) used for many of the products. Additional details on these flux products can be found in their related publications and in Friedlingstein et al. (2020).

[Figure]

**Figure 1A:** Original version of Figure 1 from manuscript, derived from three flux products (Landschützer et al. 2016, Rödenbeck et al. 2013, and Takahashi et al. 2009).

Original caption: "Distribution of the spread-based prior ocean flux uncertainty (U3) (annual average for the year 2003). It is represented here as a percentage of the prior ocean flux. The percentage shown for each grid-cell is derived from the ratio of spread-based prior ocean uncertainty divided by the prior ocean flux value at that grid cell. DJF represents the monthly average for December, January, February; MAM for March, April, May; JJA for June, July, August; SON for September, October, November. "

[Figure]

**Figure 1B**: Revised version of Figure 1.

Distribution of the spread-based prior ocean flux uncertainty (year 2003) calculated from the following 8 flux air-sea $CO_2$ flux products: (1) Denvil-Sommer et al. 2019 (product LSCE-FFNN-v1); (2) Iida et al. 2015 (JMA); (3) Zeng et al. 2015 (NIES); (4) Gregor et al. 2019 (CSIR-ML6); (5) Chau et al. 2020 (CMEMS); (6) Watson et al. 2020; (7) Landschützer et al. 2016; (8) Rödenbeck et al. 2013.

*Reviewer 1: Additionally, if Landschützer's product is used as the prior for the GCL inverse method, is it fair to use it as an independent comparison? If the data assimilation method is trying to "fit" or "correct" the GCL inverse model to the $pCO_2$ from that product then I would not consider it an independent comparison.*

**Response**: In the GEOSChem-LETKF formulation employed in this study, the data assimilated (that the flux estimates are "corrected" by) are the atmospheric observations of $CO_2$ (described in section 2.4 of our manuscript, and represented by $y$ in the LETKF model equations of the Appendix). The posterior estimates of surface $CO_2$ fluxes (specified via the analysis state equations A6 and A12) are dependent on the differences between the atmospheric observations ($y$) and the transport model derived atmospheric concentrations ($Hx^b$), and the atmospheric observations. The posterior flux estimates also do depend on the prior flux estimates (here Landschutzer et al. 2016), which are represented via the specification of the background state $x^b$, however we do not consider the Landschutzer et al. (2016) product as an independent product for comparison. One aim of the comparison of the GEOSChem-LETKF posterior fluxes to the Landschutzer product is to assess how much the posterior flux estimates vary from the prior specification following assimilation of the atmospheric $CO_2$ measurements. For example, in section 3.1, Table 1, we compare the representation of prior and posterior fluxes for three separate representations of the prior ocean flux (namely Landschützer et al. 2016,

Rödenbeck et al. 2013, and Takahashi et al. 2009). In our discussion of the multi-year analyses (2000-2017) of section 3.2, we have noted in the caption of Figure 2 that the prior flux used is that of Landschutzer et al. (2016). We will clarify this in more detail in the discussion of section 3.2 as well.

*Reviewer 1: Section 2.5 could use more discussion/explanation. To my understanding, while the inverse model itself is not new, this method of specifying prior $CO_2$ fluxes and using them to create more robust flux uncertainties seems to be a major improvement described in this manuscript. I'd be keen to see more explanation and discussion on that in this section.*

**Response:**

To address this reviewer comment we will add the following discussion to section 2.5. This discussion will also include the extended list of $pCO_2$-based flux products used in the updated analyses of this Response to Reviewers:

"Many previous atmospheric inverse estimates of air-sea carbon fluxes have employed relatively simple characterizations of the prior ocean flux uncertainty, for example, based on a fixed proportion of the grid-scale or regional prior flux (Nassar et al. 2011, Liu et al. 2016, Feng et al. 2016). The prior ocean flux distributions employed in atmospheric inversions are frequently derived from interpolations of the surface ocean $pCO_2$ database (e.g., SOCAT, Bakker et al. 2016) in combination with parameterizations of air-sea gas-exchange. Uncertainties in the derived products stem from uncertainties in the input data (e.g., density of measurements), interpolation methods, and gas-transfer parameterizations (Landschutzer et al. 2013). However, some ocean regions, the North Atlantic in particular, have a higher density of $pCO_2$ measurements and more consistent flux estimates from $pCO_2$-based products (Schuster et al. 2013, Landschutzer et al. 2013). Here we exploit the recent expansion of $pCO_2$-based ocean flux products to outline a new specification of ocean prior flux uncertainty. The uncertainty scheme uses a diagnostic derived from the variation among a set of ocean-atmosphere carbon flux products to provide an improved representation of prior flux uncertainty. This scheme specifies lower uncertainty levels where alternative prior flux representations are in accord (e.g., when well-constrained by availability of surface $pCO_2$ measurements), and higher uncertainty levels where the prior flux distributions differ significantly (typically in under-sampled regions or those of significant flux variability). The ocean flux products incorporated in our analysis include the following: Landschutzer et al. 2016, Rodenbeck et al. 2013, Denvil-Sommer et al. 2019, Iida et al. 2015, Zeng et al. 2015, Gregor et al. 2019, Chau et al. 2020, and Watson et al. 2020".

*Reviewer 1: Lastly, I think the title could be more descriptive of the actual work you are presenting. Specifically mentioning inverse models or uncertainty or a comparison between approaches.*

**Response:** We propose to change the title of the study to "Variability of North Atlantic $CO_2$ fluxes for the 2000–2017 period estimated from a local ensemble transform Kalman filter (GEOSChem-LETKF)"

*Reviewer 1: Minor comments*

*Reviewer 1: The end of the introduction could improve from a motivation statement. Why do this work? Who will use this? How will it impact the community and what are the broader impacts? Clearly your improvements on the uncertainty estimates would be beneficial to the community as a whole so make that case more clearly.*
**Response:** To address this comment we propose to add additional discussion to the introduction of the manuscript (following line 59).

"Atmospheric inverse analyses providing estimates of global-scale surface $CO_2$ fluxes have previously relied on one or two individual ocean carbon products (e.g., Takahashi et al. 2009, Landschutzer et al. 2013) for specification of prior fluxes. Following recent updates the surface ocean $pCO_2$ database SOCATv2020 (Bakker et al. 2016, 2020), now includes over 28 million surface ocean carbon measurements. This has provided a valuable resource for bottom-up estimates of ocean-atmosphere $CO_2$ fluxes, and the SOCAT measurements have been interpolated using a variety of mapping methods by a range of recent studies to provide ocean flux products. The increased range of products available (e.g., those reported in the recent Global Carbon Budget (Friedlingstein et al. 2020), and the 8 interannually varying air-sea flux products collected for this study, provide a valuable opportunity to develop an improved representation of ocean variability and more robust characterization of the uncertainties associated with ocean carbon fluxes. In this study we provide a new method of characterizing the prior ocean flux uncertainty used for atmospheric inverse analyses based on the ensemble spread of the multiple ocean flux products."

*Reviewer 1: Why is year 2003 selected for sensitivity tests on the prior flux uncertainty? Is three years of spin up sufficient? Is 2003 an anomalous year at all? With the dynamics at play in the North Atlantic basin it is important to consider how the selection of one year of focus can influence your analysis.*
**Response:** We add more information about the selection of year 2003 in section 2.3:
We conducted a 3-year model spin-up, starting from January 1st, 2000; the length of spin-up was determined by recommendations on the duration required for stabilization of tropospheric $CO_2$ gradients (e.g., Gurney et al. 2002), and following methods used for previous GEOSChem $CO_2$ analyses (e.g., Nassar et al. 2010). The year 2003 was selected for sensitivity tests as the first viable year following spin-up. Analyses of inter-annual variability in Atlantic $CO_2$ (e.g., Landschutzer et al. 2013; Schuster et al. 2013) do not find 2003 to be an anomalous year for regional ocean fluxes.

*Reviewer 1: It could be clarified that when you move on from Section 3.1 you will only be using the U3 approach to specify uncertainty. Additionally, the same for your selection of Landschützer et al. 2017 as the prior for the GCL model as you transition to analysis in Section 3.2.*
**Response:** We will add clarification to section 3.2 to specify the prior flux and uncertainty scheme used.

**Reviewer 1:** *Figure 2: I find it very interesting that the CTE is so anomalous in the NA subtropics but the CT model is more anomalous in the subpolar regions. It jumps out at you from this figure and you barely notice anything else. Would be worth further discussion as to why those are so different in their mean, IAV, and decadal variability. What do these other inverse methods use as prior flux inputs?*

**Response:** A potential reason for the anomalous behaviour of the CT and CTE models in the North Atlantic could be the underlying prior flux distributions. For example, the CTE results shown in Figure 2 are derived from the CTE 2016 model version (van der Laan-Luijkx et al. 2017) and use ocean prior fluxes from the ocean model inversion of Jacobson et al. (2007); i.e., the prior fluxes are not derived from the $pCO_2$-based flux products used in many other inverse analyses. In addition, Peylin et al. (2013) have noted that significant variability in atmospheric inverse IAV estimates is a potential indicator of 'flux leakage', where significant variability of terrestrial carbon fluxes in combination with sparse atmospheric sampling can result in misattribution of carbon flux estimates between land and ocean. See additional discussion of this in our Response to Reviewer 2.

**Reviewer 1:** *Figure 3: could be cleaned up and simplified by reducing the y-label axis ticks and tick labels. Additionally, on this figure, if the trends in subplots e and f are not significant, consider making the filling color gray or something else to distinguish. They should be noted on the figure as well as in the table. Currently you only note that one of the GCL trends is significant and one is not but need to do this for all inverse models, $pCO_2$ products, and GOBMs as well.*

**Response:** We have addressed the reviewer comments on Figure 3. Changes made include: (i) changes to y-axis labels and ticks; (ii) inclusion of the 6 additional $pCO_2$-based flux products introduced in this revision and discussed above; (iii) for panels (e) and (f) emphasizing the cases where the derived trends are significant (unfilled symbols in (e) and (f) are for cases where the estimated trend is not significant). A revised version of Figure 3 is shown below.

In addition, we have also augmented the tables corresponding to Figure 3, to account for the additional $pCO_2$ based products. The revised sections of Tables 2, 3, and 4 concerning the additional $pCO_2$-based flux products are also listed below.

[Figure]

**Figure 3 (revised).** Comparison of $CO_2$ ocean flux metrics for the 2000−2017 period for North Atlantic subtropics (left panels) and subpolar regions (right panels). Metrics shown are the long term mean (panels (a) and (b)); interannual variability (IAV) (panels (c) and (d)); and long term trend (panels (e) and (f)). The GCL estimates (red stars) are shown in comparison to other atmospheric inverse analyses (red symbols), surface ocean $pCO_2$ products (blue) and global ocean biogeochemistry models (GOBMs, purple). Also shown are the estimated means from each sub-group of analyses (circle symbols) with associated uncertainty (1 standard deviation). The unfilled symbols in (e) and (f) represents the trend is not significant.

| Long term mean (PgC y$^{-1}$) | | |
|---|---|---|
| NA Subtropics | NA Subpolar | |
| **Surface ocean pCO₂-based flux products** | | |
| -0.263 | -0.23 | pCO₂La (Landschutzer et al. 2016) |
| -0.284 | -0.252 | pCO₂Ro (Rodenbeck et al. 2013) |
| -0.264 | -0.208 | CMEMS (Chau et al. 2020) |
| -0.302 | -0.248 | CSIR (Gregor et al. 2019) |
| -0.295 | -0.241 | JMA (Iida et al. 2015) |
| -0.309 | -0.192 | LSCEFFNN (Denvil-Sommer et al. 2019) |
| -0.193 | -0.171 | NIES (Zeng et al. 2015) |
| -0.305 | -0.259 | Watson et al. (2020) |
| -0.277±0.077 | -0.225±0.064 | Mean of all pCO₂-based representations |

**Table 2 revised subsection** : Revised section on pCO₂-based flux products to include the six additional interannually varying flux data products.

| Interannual Variability (IAV) (PgC y$^{-1}$) | | |
|---|---|---|
| NA Subtropics | NA Subpolar | |
| **Surface ocean pCO$_2$–based flux products** | | |
| 0.038 | 0.036 | pCO$_2$La (Landschutzer et al. 2016) |
| 0.050 | 0.035 | pCO$_2$Ro (Rodenbeck et al. 2013) |
| 0.031 | 0.018 | CMEMS (Chau et al. 2020) |
| 0.035 | 0.020 | CSIR (Gregor et al. 2019) |
| 0.028 | 0.010 | JMA (Iida et al. 2015) |
| 0.037 | 0.017 | LSCEFFNN (Denvil-Sommer et al. 2019) |
| 0.029 | 0.020 | NIES (Zeng et al. 2015) |
| 0.037 | 0.040 | Watson et al. (2020) |
| 0.036±0.015 | 0.024±0.021 | Mean of all pCO$_2$-based representations |

**Table 3 revised subsection**: Revised section on pCO$_2$-based flux products to include the six additional interannually varying flux data products**.**

| Trend (PgC y$^{-1}$ decade$^{-1}$) | | |
|---|---|---|
| NA Subtropics | NA Subpolar | |
| **Surface ocean pCO$_2$-based flux products** | | |
| -0.068(S) | -0.056(S) | pCO$_2$La (Landschutzer et al. 2016) |
| -0.070(S) | -0.029 | pCO$_2$Ro (Rodenbeck et al. 2013) |
| -0.057(S) | -0.027(S) | CMEMS (Chau et al. 2020) |
| -0.063(S) | -0.034(S) | CSIR (Gregor et al. 2019) |
| -0.048(S) | -0.001 | JMA (Iida et al. 2015) |
| -0.070(S) | -0.021 | LSCEFFNN (Denvil-Sommer et al. 2019) |
| -0.057(S) | -0.037(S) | NIES (Zeng et al. 2015) |
| -0.069(S) | -0.064(S) | Watson et al. (2020) |
| -0.063±0.017 | -0.034±0.038 | Mean of all pCO$_2$-based representations |

**Table 4 revised** subsection: Revised section on pCO$_2$-based flux products to include the six additional interannually varying flux data products.

*Reviewer 1: Throughout the paper, where you mention Figure 3, please add the subplot letter so that the reader can easily navigate to which subplot you are referencing (e.g. Line 268: "Fig 3e, Table 4)". Also, in the Table 2 caption you could reference "Figure 3 a,b" rather than just "Fig 3".*

**Response:** To address the above reviewer comments, we have made the following changes to the manuscript:

Table 2 caption: changed to "The metrics listed in this table are plotted in Fig. 3 a, b"

Table 3 caption: changed to "The metrics listed in this table are plotted in Fig.3 c, d"
Table 4 caption: changed to "The metrics listed in this table are plotted in Fig. 3 e, f"
Line 225 changed to "Figure 3a"
Line 236 changed to "Figure 3 c, d"
Line 269 changed to "Figure 3 e"

***Reviewer 1:*** *Section 3.2.2. should include further discussion and references explaining why the GOBMs have such low IAV as compared to the other products and inverse models.*
**Response:** In response to this reviewer comment we will add the following discussion to section 3.2.2:
"Recent synthesis studies of global ocean carbon fluxes have noted that GOBMs underestimate the magnitude of IAV in comparison to estimates from $pCO_2$-based mappings and inverse analyses (DeVries et al.2019, Hauck et al.2020). An important driver of IAV is the variability in biological carbon export; the lower variability observed in the GOBMs could result from opposing changes in biological vs. circulation impacts on carbon export, which potentially reduces the sensitivity of the GOBM air-sea carbon fluxes to climate variability (Landschutzer et al. 2013, DeVries et al. 2019)."

***Reviewer 1***: *Line 225: Your first comparison is to Schuster et al. 2013 but that work is looking at a very different time period. While it can still be referenced and mentioned, highlighting comparisons that focus on the same decades of analysis is more appropriate.*
**Response:** We have rewritten this discussion to focus on the estimates from the recent $pCO_2$-based products and GOBMs, as summarised in our revised versions of Figure 3a and Table 2. We will note that the GCL posterior flux estimate of $-0.26\pm0.04$ PgC $y^{-1}$ is in agreement with estimates from the extended set of $pCO_2$-based products (mean of $-0.277\pm0.077$PgC $y^{-1}$), and from GOBMs (mean of $-0.246\pm0.076$ PgC $y^{-1}$).

***Reviewer 1:*** *Your summary statement on beginning on Line 303 could be expanded on. How is it more "robust"? Is it just smaller uncertainties and significant trends? Perhaps tie in reference to Figure 1 to discuss further.*
**Response:** We have rewritten this summary statement (Line 303) to reference our specific findings, as follows:
"Air-sea $CO_2$ flux estimates and associated metrics derived from our GCL analyses are generally more robust for the NA subtropics (which display smaller levels of spread-based prior flux uncertainty, Figure 1) than for the NA subpolar region. The posterior flux estimates for the NA sub-tropics are characterized by smaller uncertainty bounds (Figure 2), and the derivation of statistically significant trends for the long-term analyses (Table 4). "
*Reviewer 2: "The authors have studied the CO₂ annual fluxes in the North Atlantic during an 18-yr period with an atmospheric inverse modelling approach. They show some agreement with other estimates and present a sensitivity study with respect to the prior ocean flux constraint. The topic is obviously of great interest but the actual paper is rather deceiving, with little scientific depth. I am listing here important questions that are fully in the paper scope but that seem to be left open. How significant are the presented sensitivity tests for the inversion community? Despite a subsection and an appendix devoted to it, the description of the data assimilation system is unclear on what matters in practice. My interpretation of l. 95 is that the elementary assimilation window of the LETKF is of four weeks, a period which is too short (given mixing time scales in the atmosphere) to allow a clear distinction between the uncertainty in the prior initial state of atmospheric CO₂ and the uncertainty in the prior surface fluxes, when assimilating atmospheric measurements. The authors should therefore not separate the two. However, Incidentally, in the legend of Eq. A1 in the appendix, we understand that the uncertainty in the initial state of atmospheric CO₂ has been neglected. This rough simplification makes it hard to interpret B, officially the flux covariance matrix, in these terms.*

**Response:** We thank the reviewer for their comments on our manuscript. To respond to the reviewer comments above, we have added additional detail on the LETKF data assimilation system to section 2.5 and to the Appendix, to clarify our description of the system. Here we provide a summary in response to the reviewer's specific questions on the assimilation window. Additional detail on the specification of prior flux uncertainties is provided in our responses to Reviewer 1 above, and in further responses to Reviewer 2 (below).

The GEOSChem-LETKF data assimilation system employed in this study provides estimates of surface CO₂ fluxes at grid-scale resolution (2.5º x 2 º, in this instance) for the period 2000–2017. Our methods follow the implementation of the LETKF system by Liu et al. (2019), who have extended the previous carbon-data assimilation system of Kang et al. (2011, 2012). The study of Kang et al. (2011) assimilated meteorological data and atmospheric CO₂ concentrations to provide estimated atmospheric CO₂ concentrations as part of the state estimate. Kang et al. (2012) extended this method to also provide estimates of surface carbon fluxes. Both these LETKF studies assimilated meteorological data and atmospheric CO₂ concentrations and employed a short assimilation window of 6 hours in order to maintain linear behaviour of the ensemble perturbations (Kang et al. 2011, 2012). In addition, Kang et al. (2012) also tested longer assimilation windows (up to 3 weeks) for LETKF formulations that assimilated atmospheric CO₂ concentrations alone (eliminating the assimilation of the meteorological data). The LETKF system of Liu et al. (2019) extended the Kang et al. (2011, 2012) analyses by incorporating the GEOSChem atmospheric model as the forecast model, along with its representation of surface CO₂ fluxes which provide the prior flux specification for the forecast step. However, Liu et al. (2019) assimilate only atmospheric CO₂ measurements (i.e., no assimilation of meteorological measurements), and extend the assimilation window to 7 days; the duration of the assimilation window is selected to maximize the correlation between observations and surface fluxes. The GEOSChem-LETKF system employed in our study follows the Liu et al. (2019) formulation; atmospheric CO₂ measurements are assimilated at 7 day timescales, with the LETKF analysis step providing updates of the surface fluxes and associated uncertainties required as initial conditions for the next weekly forecast step. We report monthly flux estimates following four assimilation cycles.

***Reviewer 2***: *Similarly, the authors do not discuss spatial correlations in the prior errors, leaving the impression that they have neglected them as well. How credible is this hypothesis, e.g., among the prior ocean flux products tested here?*

**Response:** An overview of the prior ocean $CO_2$ fluxes and uncertainties is presented in section 2.5 of the manuscript. We will extend this section to include clarification on the spatial correlations in prior flux uncertainties, to include the following discussion:

"In this study we account for spatial correlations in the prior ocean fluxes, by inclusion of off-diagonal elements in the background error covariance matrix $P^b$ (Equation A3). We follow the recommendations of Jones et al. (2012) on autocorrelation length scales in the surface ocean. That study derived spatial autocorrelation functions for air-sea fluxes from an analysis of the surface ocean $pCO_2$ database reported in Takahashi et al. (2009), combined with a gas-exchange parameterization. We currently do not account for spatial correlation in land-fluxes, but will investigate this in future analyses."

***Reviewer 2***: *How are the ocean flux results presented here affected by the leakage from the land fluxes noted in l. 48? The statement in l. 156 suggests there is none of significance, but without any justification.*

**Response:**  Peylin et al. (2013) note the potential for flux 'leakage' from the land as an influence on estimates of northern hemisphere ocean flux estimates, whereby the larger magnitude of the variability of terrestrial carbon fluxes in combination with sparse atmospheric sampling can result in misattribution of carbon flux estimates between land and ocean. To assess the significance of flux leakage in our analyses, we calculate estimates of the diagnostic recommended by Peylin et al. (2013) (i.e., the correlation between the annual total land and total ocean fluxes) for the Northern Hemisphere as a whole (Equator to 90N), and also by latitudinal region.  We do not find significant leakage for the Northern Hemisphere as a whole (correlation coefficient of ~ 0.3), or for the Northern Hemisphere sub-tropics (correlation coefficient of ~0.05), however the diagnostic indicates increased potential for flux leakage in the Northern Hemisphere sub-polar region (correlation coefficient ~0.51). This analysis was conducted for the period 2000-2017 and we will add this discussion to section 3.2 on the results of multi-year analyses.

***Reviewer 2:*** *Additionally, a number of points of various important need clarification:*
*• L. 27: the 20% value is rather artificial given the fact that the global ocean uptake is made of both sources and sinks.*

**Response:** In our original manuscript the 20% value for North Atlantic carbon uptake was reported as it represented a percentage of global net carbon uptake by the ocean, as calculated for example, by such synthesis projects as the Global Carbon Budget (Friedlingstein et al. 2020). In our revised manuscript we will update this section to include more recent findings, and also to include updated calculations of North Atlantic carbon uptake derived from our synthesis of the 8 $pCO_2$-based flux products (see comments in Response to Reviewer 1, and the updated section of Table 2 included in the Response).

Our updates to this section in the Introduction will include the following information:

"Recent estimates of net air-sea $CO_2$ fluxes derived from sea surface partial pressure $CO_2$ measurements ($pCO_2$) indicate net annual uptake for the North Atlantic over the past decade (2009-2018) with a range of 0.35–0.55 PgC $y^{-1}$ (Landschutzer et al. 2016; Rodenbeck et al. 2013; Zeng et al. 2015; Watson et al. 2020), and equivalent to about 15%-35% of the global

net ocean carbon ocean sink reported for this period (i.e., -2±0.5 PgC y$^{-1}$; Friedlingstein et al. 2020)."

*Reviewer 2: L. 57-8: bad example; the studies mentioned here are not for the same year and therefore should not use the same uncertainty budget for a frozen prior flux distribution anyway, given existing trends in the real fluxes.*

**Response:** Our aim in using these examples was to illustrate the relatively simple characterizations of prior flux uncertainty in previous inverse assessments of ocean carbon fluxes, due to the limited information available at the time on variability of ocean-atmosphere carbon fluxes. We have rewritten this section to clarify this message. Our revised version of this section will include the following discussion:

"Several global inverse model assessments of the past decade have employed the climatological data for ocean-atmosphere $CO_2$ fluxes from Takahashi et al. (2009) to specify prior ocean fluxes (e.g., Nassar et al. 2011, Feng et al. 2016, Deng et al. 2016). In view of the limited information available on the temporal and spatial variability of ocean carbon fluxes from the climatological ocean database, these inverse analyses have adopted different approaches to the specification of prior uncertainty for ocean fluxes, ranging from uncertainties derived from a separate ocean model inversion (in the case of Nassar et al. 2011), to a specified percentage of the prior flux magnitude (Feng et al. 2016). "

*Reviewer 2: L. 91: this is Appendix A, not A1.*
**Response:** We have corrected it to Appendix A.

*Reviewer 2: L. 121: what is the rationale behind the 60% and 120% values? The authors should relate them to their knowledge of the quality of their prior fluxes, while they make it look arbitrary (except if indeed matrix B is just an ensemble of tuning factors and not an error covariance matrix; see above).*

**Response:** The selection of the prior uncertainty levels used in the sensitivity analyses of section 3.1 was based on the range of variability seen for the individual prior flux distributions (Takahashi et al. 2009; Landschutzer et al. 2016; and Rödenbeck et al. 2013) for the sub-regions of the North Atlantic. These ranged from average levels of ~60% for the sub-tropical North Atlantic to levels > 120% for the sub-polar North Atlantic, hence we selected a level of U1:60% to characterize the lower sensitivity case, and U2:120% for the higher case.

*Reviewer 2: L. 127: what is the value of K? I get the impression that only 3 flux products are used here: no standard deviation can be estimated from just three members.*

**Response:** In our original manuscript K = 3, as we included only three prior flux distributions. In response to suggestions from Reviewer 1, we have now included six other inter-annually varying pCO$_2$-based flux products in our analyses (see discussion and revised results in our Response to Reviewer 1 above). For the revised version of Figure 1 included in our response above K = 8 (we have omitted the climatological flux distribution of Takahashi et al. 2009, in the revised figure above).

***Reviewer 2***: *L. 140-1: why would the three prior ocean flux distributions have the same uncertainty statistics?*

**Response:** The uncertainty statistics of the prior ocean flux distributions will be dependent on the uncertainties associated with the respective inputs and methods of constructing the flux products. Ocean-atmosphere carbon flux products derived from surface ocean $pCO_2$ measurements are generally subject to two main sources of uncertainty: (i) in the specification of the surface $CO_2$ partial pressure difference across the air-sea interface, and (ii) in the specification of the gas-exchange coefficient used to derive fluxes (e.g., see discussion of Landschutzer et al. 2013; Watson et al. 2020). In the extended database of 8 $pCO_2$-based flux products that we present above in this Response to Reviewers, the majority of the flux products (seven of the eight) rely on the surface ocean $pCO_2$ data of the SOCAT database (Bakker et al. 2016, 2020). These flux products will be subject to similar uncertainties associated with data coverage in different ocean regions, although the uncertainties due to differences among surface interpolation methods may vary. We will add discussion of these sources of uncertainty to section 3.1.

***Reviewer 2***: *L. 170: flexibility is not the question. The question is about well modelling the prior uncertainty.*

**Response:** We have rewritten this section to better describe the aim of using the spread-based uncertainty-scheme. The section will include the following discussion:

"Several previous atmospheric inverse estimates of surface ocean $CO_2$ fluxes have relied on a relatively simple characterization of prior flux uncertainty, for example, based on a fixed proportion of the grid-scale or regional prior flux. The uncertainty scheme outlined here uses a diagnostic derived from the variation among a set of ocean-atmosphere carbon flux products to provide an improved representation of prior flux uncertainty. This scheme specifies lower uncertainty levels where alternative prior flux representations are in accord (e.g., when well-constrained by availability of surface $pCO_2$ measurements), and higher uncertainty levels where the prior flux distributions differ significantly (typically in under-sampled regions or those of significant flux variability)."

***Reviewer 2***: *Table 2: if the numbers behind plus/minus signs for the mean values across studies are standard deviations, how can they have been computed on 6, 3, or even 2 members only?*

**Response:** We will report the uncertainty ranges in Table 2 using statistics of small samples (e.g., Hollander et al. 2014). We note that the number of $pCO_2$-based flux products in our revised study has now increased to eight (see revised version of Table 2) in our Response to Reviewer 1. The uncertainty ranges reported in the revised version for Table 2 represent the 95% confidence interval based on the Student's t distribution.

***Reviewer 2***: *L. 339: what is the value of L?*

**Response:** The parameter L represents the horizontal localization radius, and is set to 2000 km for this study, following Liu et al. (2016). The localization radius is used in the LETKF in a latitude-dependent weighting function which characterizes the spatial scale of the region within which atmospheric $CO_2$ observations are assimilated at each gridpoint (Miyoshi et al. 2007). We will add further clarification of this to section 2.2 and to the Appendix.

**References for Reviewer 1 Responses:**

[revised manuscript text omitted]